# Unsupervised Model Selection for Variational Disentangled Representation Learning

**Sunny Duan**[*]
DeepMind
sunnyd@google.com

**Loic Matthey**
DeepMind
lmatthey@google.com

**Andre Saraiva**
DeepMind
andresnds@google.com

**Nick Watters**
DeepMind
nwatters@google.com

**Chris Burgess**
DeepMind
cpburgess@google.com

**Alexander Lerchner**
DeepMind
lerchner@google.com

**Irina Higgins**[*]
DeepMind
irinah@google.com

## Abstract

Disentangled representations have recently been shown to improve fairness, data efficiency and generalisation in simple supervised and reinforcement learning tasks. To extend the benefits of disentangled representations to more complex domains and practical applications, it is important to enable hyperparameter tuning and model selection of existing unsupervised approaches without requiring access to ground truth attribute labels, which are not available for most datasets. This paper addresses this problem by introducing a simple yet robust and reliable method for unsupervised disentangled model selection. Our approach, Unsupervised Disentanglement Ranking (UDR)[1], leverages the recent theoretical results that explain why variational autoencoders disentangle (Rolinek et al., 2019), to quantify the quality of disentanglement by performing pairwise comparisons between trained model representations. We show that our approach performs comparably to the existing supervised alternatives across 5400 models from six state of the art unsupervised disentangled representation learning model classes. Furthermore, we show that the ranking produced by our approach correlates well with the final task performance on two different domains.

## 1 Introduction

*Happy families are all alike; every unhappy family is unhappy in its own way. —*

Leo Tolstoy, Anna Karenina

Despite the success of deep learning in the recent years (Hu et al., 2018; Espeholt et al., 2018; Silver et al., 2018; Lample et al., 2018; Hessel et al., 2017; Oord et al., 2016), the majority of state of the art approaches are still missing many basic yet important properties, such as fairness, data efficient learning, strong generalisation beyond the training data distribution, or the ability to transfer knowledge between tasks (Lake et al., 2016; Garnelo et al., 2016; Marcus, 2018). The idea that a good representation can help with such shortcomings is not new, and recently a number of papers have demonstrated that models with *disentangled* representations show improvements in terms of these shortcomings (Higgins et al., 2017b; 2018b; Achille et al., 2018; Steenbrugge et al., 2018; Nair et al., 2018; Laversanne-Finot et al., 2018; van Steenkiste et al., 2019; Locatello et al., 2019). A common intuitive way to think about disentangled representations is that it should reflect the compositional

---

[*]Equal contribution.
[1]We have released the code for our method as part of `disentanglement_lib`

Figure 1: Latent traversals for one of the best and worst ranked trained $\beta$-VAE models using the Unsupervised Disentanglement Ranking (UDR$_L$) method on the 3D Cars dataset. For each seed image we fix all latents $z_i$ to the inferred value, then vary the value of one latent at a time to visualise its effect on the reconstructions. The high scoring model (left 3 blocks) appears well disentangled, since individual latents have consistent semantic meaning across seeds. The low scoring model (right block) is highly entangled, since the latent traversals are not easily interpretable.

structure of the world. For example, to describe an object we often use words pertaining to its colour, position, shape and size. We can use different words to describe these properties because they relate to independent factors of variation in our world, i.e. properties which can be compositionally recombined. Hence a disentangled representation of objects should reflect this by factorising into dimensions which correspond to those properties (Bengio et al., 2013; Higgins et al., 2018a).

The ability to automatically discover the compositional factors of complex real datasets can be of great importance in many practical applications of machine learning and data science. However, it is important to be able to learn such representations in an unsupervised manner, since most interesting datasets do not have their generative factors fully labelled. For a long time scalable unsupervised disentangled representation learning was impossible, until recently a new class of models based on Variational Autoencoders (VAEs) (Kingma & Welling, 2014; Rezende et al., 2014) was developed. These approaches (Higgins et al., 2017a; Burgess et al., 2017; Chen et al., 2018; Kumar et al., 2017; Kim & Mnih, 2018) scale reasonably well and are the current state of the art in unsupervised disentangled representation learning. However, so far the benefits of these techniques have not been widely exploited because of two major shortcomings: First, the quality of the achieved disentangling is sensitive to the choice of hyperparameters, however, model selection is currently impossible without having access to the ground truth generative process and/or attribute labels, which are required by all the currently existing disentanglement metrics (Higgins et al., 2017a; Kim & Mnih, 2018; Chen et al., 2018; Eastwood & Williams, 2018; Ridgeway & Mozer, 2018). Second, even if one could apply any of the existing disentanglement metrics for model selection, the scores produced by these metrics can vary a lot even for models with the same hyperparameters and trained on the same data (Locatello et al., 2018). While a lot of this variance is explained by the actual quality of the learnt representations, some of it is introduced by the metrics themselves. In particular, all of the existing supervised disentanglement metrics assume a single "canonical" factorisation of the generative factors, any deviation from which is penalised. Such a "canonical" factorisation, however, is not chosen in a principled manner. Indeed, for the majority of datasets, apart from the simplest ones, multiple equally valid disentangled representations may be possible (see Higgins et al. (2018a) for a discussion). For example, the intuitive way that humans reason about colour is in terms of hue and saturation. However, colour may also be represented in RGB, YUV, HSV, HSL, CIELAB. Any of the above representations are as valid as each other, yet only one of them is allowed to be "canonical" by the supervised metrics. Hence, a model that learns to represent colour in a subspace aligned with HSV will be penalised by a supervised metric which assumes that the canonical disentangled representation of colour should be in RGB. This is despite the fact that both representations are equal in terms of preserving the compositional property at the core of what makes disentangled representations useful (Higgins et al., 2018a). Hence, the field finds itself in a predicament. From one point of view, there exists a set of approaches capable of reasonably scalable unsupervised disentangled representation learning. On the other hand, these models are hard to use in practice, because there is no easy way to do a hyperparameter search and model selection.

This paper attempts to bridge this gap. We propose a simple yet effective method for unsupervised model selection for the class of current state-of-the-art VAE-based unsupervised disentangled representation learning methods. Our approach, Unsupervised Disentanglement Ranking (UDR), leverages the recent

theoretical results that explain why variational autoencoders disentangle (Rolinek et al., 2019), to quantify the quality of disentanglement by performing pairwise comparisons between trained model representations. We evaluate the validity of our unsupervised model selection metric against the four best existing supervised alternatives reported in the large scale study by Locatello et al. (2018): the $\beta$-VAE metric (Higgins et al., 2017a), the FactorVAE metric (Kim & Mnih, 2018), Mutual Information Gap (MIG) (Chen et al., 2018) and DCI Disentanglement scores (Eastwood & Williams, 2018). We do so for all existing state of the art disentangled representation learning approaches: $\beta$-VAE (Higgins et al., 2017a), CCI-VAE (Burgess et al., 2017), FactorVAE (Kim & Mnih, 2018), TC-VAE (Chen et al., 2018) and two versions of DIP-VAE (Kumar et al., 2017). We validate our proposed method on two datasets with fully known generative processes commonly used to evaluate the quality of disentangled representations: dSprites (Matthey et al., 2017) and 3D Shapes (Burgess & Kim, 2018), and show that our unsupervised model selection method is able to match the supervised baselines in terms of guiding a hyperparameter search and picking the most disentangled trained models both quantitatively and qualitatively. We also apply our approach to the 3D Cars dataset (Reed et al., 2014), where the full set of ground truth attribute labels is not available, and confirm through visual inspection that the ranking produced by our method is meaningful (Fig. 1). Overall we evaluate 6 different model classes, with 6 separate hyperparameter settings and 50 seeds on 3 separate datasets, totalling 5400 models and show that our method is both accurate and consistent across models and datasets. Finally, we validate that the model ranking produced by our approach correlates well with the final task performance on two recently reported tasks: a classification fairness task (Locatello et al., 2019) and a model-based reinforcement learning (RL) task (Watters et al., 2019). Indeed, on the former our approach outperformed the reported supervised baseline scores.

## 2    OPERATIONAL DEFINITION OF DISENTANGLING

Given a dataset of observations $X = \{\boldsymbol{x}_1,...,\boldsymbol{x}_N\}$, we assume that there exist a number of plausible generative processes $g_i$ that produce the observations from a small set of corresponding $K_i$ independent generative factors $\boldsymbol{c}_i$. For each choice of $i$, $g : \boldsymbol{c}_n \mapsto \boldsymbol{x}_n$, where $p(\boldsymbol{c}_n) = \prod_{j=1}^{K} p(c_n^j)$. For example, a dataset containing images of an object, which can be of a particular shape and colour, and which can be in a particular vertical and horizontal positions, may be created by a generative process that assumes a ground truth disentangled factorisation into *shape* x *colour* x *position*, or *shape* x *hue* x *saturation* x *position X* x *position Y*. We operationalise a model as having learnt a disentangled representation, if it learns to invert of one of the generative processes $g_i$ and recover a latent representation $\boldsymbol{z} \in \mathbb{R}^L$, so that it best explains the observed data $p(\boldsymbol{z}, \boldsymbol{x}) \approx p(\boldsymbol{c}_i, \boldsymbol{x})$, and factorises the same way as the corresponding data generative factors $\boldsymbol{c}_i$. The choice of the generative process can be determined by the interaction between the model class and the observed data distribution $p(\boldsymbol{x})$, as discussed next in Sec. 3.1.

## 3    VARIATIONAL UNSUPERVISED DISENTANGLING

The current state of the art approaches to unsupervised disentangled representation learning are based on the Variational Autoencoder (VAE) framework (Rezende et al., 2014; Kingma & Welling, 2014). VAEs attempt to estimate the lower bound on the joint distribution of the data and the latent factors $p(\boldsymbol{x}, \boldsymbol{z})$ by optimising the following objective:

$$\mathcal{L}_{VAE} = \mathbb{E}_{p(\boldsymbol{x})}[\, \mathbb{E}_{q_\phi(\boldsymbol{z}|\boldsymbol{x})}[\log p_\theta(\boldsymbol{x}|\boldsymbol{z})] - KL(q_\phi(\boldsymbol{z}|\boldsymbol{x}) \,||\, p(\boldsymbol{z}))\,] \qquad (1)$$

where, in the usual case, the prior $p(\boldsymbol{z})$ is chosen to be an isotropic unit Gaussian. In order to encourage disentangling, different approaches decompose the objective in Eq. 1 into various terms and change their relative weighting. In this paper we will consider six state of the art approaches to unsupervised disentangled representation learning that can be grouped into three broad classes based on how they modify the objective in Eq. 1: 1) $\beta$-VAE (Higgins et al., 2017a) and CCI-VAE (Burgess et al., 2017) upweight the KL term; 2) FactorVAE (Kim & Mnih, 2018) and TC-VAE (Chen et al., 2018) introduce a total correlation penalty; and 3) two different implementations of DIP-VAE (-I and -II) (Kumar et al., 2017) penalise the deviation of the the marginal posterior from a factorised prior (see Sec. A.4.1 in Supplementary Material for details).

## 3.1 WHY DO VAEs DISENTANGLE?

In order to understand the reasoning behind our proposed unsupervised disentangled model selection method, it is first important to understand why VAEs disentangle. The objective in Eq. 1 does not in itself encourage disentangling, as discussed in Rolinek et al. (2019) and Locatello et al. (2018). Indeed, any rotationally invariant prior makes disentangled representations learnt in an unsupervised setting unidentifiable when optimising Eq. 1. This theoretical result is not surprising and has been known for a while in the ICA literature (Comon, 1994), however what is surprising is that disentangling VAEs appear to work in practice. The question of what makes VAEs disentangle was answered by Rolinek et al. (2019), who were able to show that it is the peculiarities of the VAE implementation choices that allow disentangling to emerge (see also discussion in Burgess et al. (2017); Mathieu et al. (2019)). In particular, the interactions between the reconstruction objective (the first term in Eq. 1) and the enhanced pressure to match a diagonal prior created by the modified objectives of the disentangling VAEs, force the decoder to approximate PCA-like behaviour locally around the data samples. Rolinek et al. (2019) demonstrated that during training VAEs often enter the so-called "polarised regime", where many of the latent dimensions of the posterior are effectively switched off by being reduced to the prior $q_\phi(z_j) = p(z_j)$ (this behaviour is often further encouraged by the extra disentangling terms added to the ELBO). When trained in such a regime, a linear approximation of the Jacobian of the decoder around a data sample $\boldsymbol{x}_i$, $J_i = \frac{\partial Dec_\theta(\mu_\phi(\boldsymbol{x}_i))}{\partial \mu_\phi(\boldsymbol{x}_i)}$, is forced to have orthogonal columns, and hence to separate the generative factors based on the amount of reconstruction variance they induce. Given that the transformations induced by different generative factors will typically have different effects on the pixel space (e.g. changing the position of a sprite will typically affect more pixels than changing its size), such local orthogonalisation of the decoder induces an identifiable disentangled latent space for each particular dataset. An equivalent statement is that for a well disentangled VAE, the SVD decomposition $J = U\Sigma V^\top$ of the Jacobian $J$ calculated as above, results in a trivial $V$, which is a signed permutation matrix.

## 4 UNSUPERVISED DISENTANGLED MODEL SELECTION

We now describe how the insights from Sec. 3.1 motivate the development of our proposed Unsupervised Disentanglement Ranking (UDR) method. Our method relies on the assumption that for a particular dataset and a VAE-based unsupervised disentangled representation learning model class, disentangled representations are all alike, while every entangled representation is entangled in its own way, to rephrase Tolstoy. We justify this assumption next.

**Disentangled representations are similar** According to Rolinek et al. (2019) for a given non-adversarial dataset a disentangling VAE will likely keep converging to the same disentangled representation (up to permutation and sign inverse). Note that this representation will correspond to a single plausible disentangled generative process $g_i$ using the notation we introduced in Sec. 2. This is because any two different disentangled representations $\boldsymbol{z}_a$ and $\boldsymbol{z}_b$ learnt by a VAE-based model will only differ in terms of the corresponding signed permutation matrices $V_a$ and $V_b$ of the SVD decompositions of the locally linear approximations of the Jacobians of their decoders.

**Entangled representations are different** Unfortunately the field of machine learning has little theoretical understanding of the nature and learning dynamics of internal representations in neural networks. The few pieces of research that have looked into the nature of model representations (Raghu et al., 2017; Li et al., 2016; Wang et al., 2018; Morcos et al., 2018) have been empirical rather than theoretical in nature. All of them suggest that neural networks tend to converge to different hidden representations despite being trained on the same task with the same hyperparameters and architecture and reaching similar levels of task performance. Furthermore, the theoretical analysis and the empirical demonstrations in Rolinek et al. (2019) suggest that the entangled VAEs learn representations that are different at least up to a rotation induced by a non-degenerate matrix $V$ in the SVD decomposition of the local linear approximation of the decoder Jacobian $J_i$.

The justifications presented above rely on the theoretical work of Rolinek et al. (2019), which was empirically verified only for the $\beta$-VAE. We have reasons to believe that the theory also holds for the other model classes presented in this paper, apart from DIP-VAE-I. We empirically verify that this is the case in Sec. A.10 in Supplementary Materials. Furthermore, in Sec. 5 we show that our proposed method works well in practice across all model classes, including DIP-VAE-I.

**Unsupervised Disentanglement Ranking**  Our proposed UDR method consists of four steps (illustrated in Fig. 4 in Supplementary Material):

1. Train $M = H \times S$ models, where $H$ is the number of different hyperparameter settings, and $S$ is the number of different initial model weight configurations (seeds).
2. For each trained model $i \in \{1,...,M\}$, sample without replacement $P \leq S$ other trained models with the same hyperparameters but different seeds.
3. Perform $P$ pairwise comparisons per trained model and calculate the respective $\text{UDR}_{ij}$ scores, where $i \in \{1,...,M\}$ is the model index, and $j \in \{1,...,P\}$ is its unique pairwise match from Step 2.
4. Aggregate $\text{UDR}_{ij}$ scores for each model $i$ to report the final $\text{UDR}_i = \text{avg}_j(\text{UDR}_{ij})$ scores, where $\text{avg}_j(\cdot)$ is the median over $P$ scores.

The key part of the UDR method is Step 3, where we calculate the $\text{UDR}_{ij}$ score that summarises how similar the representations of the two models $i$ and $j$ are. As per the justifications above, two latent representations $\boldsymbol{z}_i$ and $\boldsymbol{z}_j$ should be scored as highly similar if they axis align with each other up to *permutation* (the same ground truth factor $c_k$ may be encoded by different latent dimensions within the two models, $z_{i,a}$ and $z_{j,b}$ where $a \neq b$), *sign inverse* (the two models may learn to encode the values of the generative factor in the opposite order compared to each other, $z_{i,a} = -z_{j,b}$), and *subsetting* (one model may learn a subset of the factors that the other model has learnt if the relevant disentangling hyperparameters encourage a different number of latents to be switched off in the two models). In order for the UDR to be invariant to the first scenario, we propose calculating a full $L \times L$ similarity matrix $R_{ij}$ between the individual dimensions of $\boldsymbol{z}_i \in \mathbb{R}^L$ and $\boldsymbol{z}_j \in \mathbb{R}^L$ (see Fig. 5 in Supplementary Material). In order to address the second point, we take the absolute value of the similarity matrix $|R_{ij}|$. Finally, to address the third point, we divide the UDR score by the average number of informative latents discovered by the two models. Note that even though disentangling often happens when the VAEs enter the "polarised regime", where many of the latent dimensions are switched off, the rankings produced by UDR are not affected by whether the model operates in such a regime or not.

To populate the similarity matrix $R_{ij}$ we calculate each matrix element as the similarity between two vectors $\boldsymbol{z}_{i,a}$ and $\boldsymbol{z}_{j,b}$, where $\boldsymbol{z}_{i,a}$ is a response of a single latent dimension $z_a$ of model $i$ over the entire ordered dataset or a fixed number of ordered mini-batches if the former is computationally restrictive (see Sec. A.5 in Supplementary Material for details). We considered two versions of the UDR score based on the method used for calculating the vector similarity: the non-parametric $\text{UDR}_S$, using Spearman's correlation; and the parametric $\text{UDR}_L$, using Lasso regression following past work on evaluating representations (Eastwood & Williams, 2018; Li et al., 2016). In practice the Lasso regression version worked slightly better, so the remainder of the paper is restricted to $\text{UDR}_L$ (we use $\text{UDR}_L$ and UDR interchangeably to refer to this version), while $\text{UDR}_S$ is discussed in the Supplementary Materials.

Given a similarity matrix $R_{ij}$, we want to find one-to-one correspondence between all the informative latent dimensions within the chosen pair of models. Hence, we want to see at most a single strong correlation in each row and column of the similarity matrix. To that accord, we step through the matrix $R = |R_{ij}|$, one column and row at a time, looking for the strongest correlation, and weighting it by the proportional weight it has within its respective column or row. We then average all such weighted scores over all the informative row and column latents to calculate the final $\text{UDR}_{ij}$ score:

$$\frac{1}{d_a+d_b}\left[\sum_b \frac{r_a^2 \cdot I_{KL}(b)}{\sum_a R(a,b)} + \sum_a \frac{r_b^2 \cdot I_{KL}(a)}{\sum_b R(a,b)}\right] \tag{2}$$

where $r_a = \max_a R(a,b)$ and $r_b = \max_b R(a,b)$. $I_{KL}$ indicates an "informative" latent within a model and $d$ is the number of such latents: $d_a = \sum_a I_{KL}(a)$ and $d_b = \sum_b I_{KL}(b)$. We define a latent dimension as "informative" if it has learnt a latent posterior which diverges from the prior:

$$I_{KL}(a) = \begin{cases} 1 & KL(q_\phi(z_a|\boldsymbol{x}) \,||\, p(z_a)) > 0.01 \\ 0 & \text{otherwise} \end{cases} \tag{3}$$

**UDR variations**  We explored whether doing all-to-all pairwise comparisons, with models in Step 2 sampled from the set of all $M$ models rather than the subset of $S$ models with the same hyperparameters, would produce more accurate results. Additionally we investigated the effect of choosing different numbers of models $P$ for pairwise comparisons by sampling $P \sim U[5,45]$.

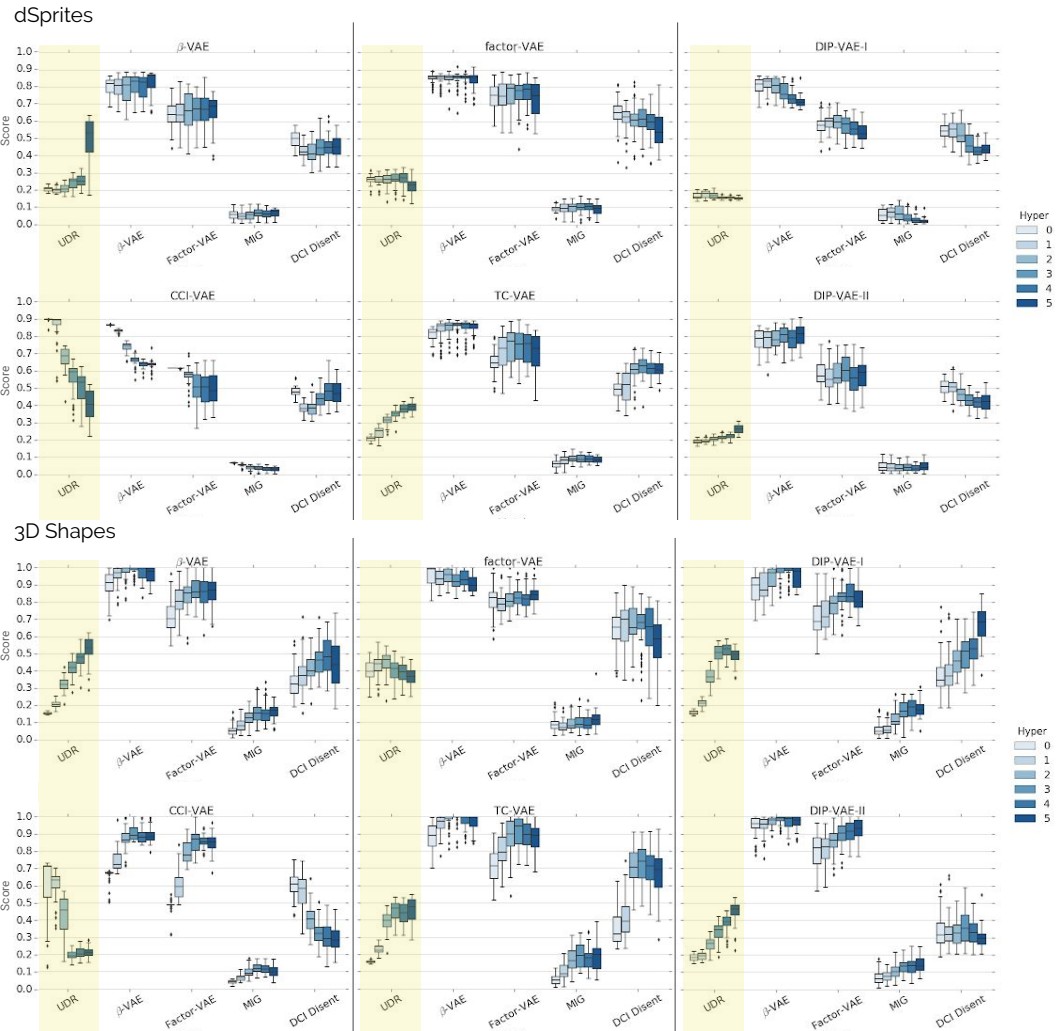

Figure 2: Hyperparameter search results for six unsupervised disentangling model classes evaluated using the unsupervised UDR and the supervised $\beta$-VAE, FactorVAE, MIG and DCI Disentangling metrics and trained on either dSprites (**top**) or 3D Shapes (**bottom**) datasets. "Hyper" corresponds to the particular hyperparameter setting considered (see Tbl. 5 in Supplementary Materials for particular values). The box and whisker plots for each hyperparameter setting are summarising the scores for 50 different model seeds. Higher median values indivate better hyperparameters. The ranking of hyperparameters tends to be similar between the different metrics, including UDR.

**UDR assumptions and limitations**    Note that our approach is aimed at the current state of the art disentangling VAEs, for which the assumptions of our metric have been demonstrated to hold (Rolinek et al., 2019). It may be applied to other model classes, however the following assumptions and limitations need to be considered:

1. **Disentangled representations produced by two models from the same class trained on the same dataset are likely to be more similar than entangled representations** – this holds for disentangling VAEs (Rolinek et al., 2019), but may not hold more broadly.

2. **Continuous, monotonic and scalar factors** – UDR assumes that these properties hold for the data generative factors and their representations. This is true for the disentangling approaches described in Sec. 3, but may not hold more generally. It is likely that UDR can be adapted to work with other kinds of generative factors (e.g. factors with special or no geometry) by exchanging the similarity calculations in Step 3 with an appropriate measure, however we leave this for future work.

| MODEL CLASS | DSPRITES | | | | 3DSHAPES | | | |
| | LASSO | | SPEARMAN | | LASSO | | SPEARMAN | |
| | HYPER | ALL-2-ALL | HYPER | ALL-2-ALL | HYPER | ALL-2-ALL | HYPER | ALL-2-ALL |
|---|---|---|---|---|---|---|---|---|
| $\beta$-VAE | 0.60 | 0.76 | 0.54 | 0.72 | 0.71 | 0.68 | 0.70 | 0.71 |
| TC-VAE | 0.40 | 0.67 | 0.37 | 0.60 | 0.81 | 0.79 | 0.81 | 0.75 |
| DIP-VAE | 0.61 | 0.69 | 0.65 | 0.72 | 0.75 | 0.74 | 0.75 | 0.78 |

Table 1: Rank correlations between MIG and different versions of UDR across two datasets and three model classes. The performance is comparable across datasets, UDR versions and model classes. See Fig. 6 in Supplementary Materials for comparisons with other supervised metrics.

3. **Herd effect** – since UDR detects disentangled representations through pairwise comparisons, the score it assigns to each individual model will depend on the nature of the other models involved in these comparisons. This means that UDR is unable to detect a single disentangled model within a hyperparameter sweep. It also means that when models are only compared within a single hyperparameter setting, individual model scores may be over/under estimated as they tend to be drawn towards the mean of the scores of the other models within a hyperparameter group. Thus, it is preferable to perform the UDR-A2A during model selection and UDR during hyperparameter selection.

4. **Explicitness bias** – UDR does not penalise models that learn a subset of the data generative factors. In fact, such models often score higher than those that learn the full set of generative factors, because the current state of the art disentangling approaches tend to trade-off the number of discovered factors for cleaner disentangling. As discussed in Sec. 2, we provide the practitioner with the ability to choose the most disentangled model per number of factors discovered by approximating this with the $d$ score in Eq. 2.

5. **Computational cost** – UDR requires training a number of seeds per hyperparameter setting and $M \times P$ pairwise comparisons per hyperparameter search, which may be computationally expensive. Saying this, training multiple seeds per hyperparameter setting is a good research practice to produce more robust results and UDR computations are highly parallelisable.

To summarise, UDR relies on a number of assumptions and has certain limitations that we hope to relax in future work. However, it offers improvements over the existing supervised metrics. Apart from being the only method that does not rely on supervised attribute labels, its scores are often more representative of the true disentanglement quality (e.g. see Fig. 3 and Fig. 9 in Supplementary Materials), and it does not assume a single "canonical" disentangled factorisation per dataset. Hence, we believe that UDR can be a useful method for unlocking the power of unsupervised disentangled representation learning to real-life practical applications, at least in the near future.

## 5 EXPERIMENTS

Our hope was to develop a method for unsupervised disentangled model selection with the following properties: it should 1) help with hyperparameter tuning by producing an aggregate score that can be used to guide evolutionary or Bayesian methods (Jaderberg et al., 2018; Snoek et al., 2012; Thornton et al., 2012; Bergstra et al., 2011; Hutter et al., 2011; Miikkulainen et al., 2017); 2) rank individual trained models based on their disentanglement quality; 3) correlate well with final task performance. In this section we evaluate our proposed UDR against these qualities. For the reported experiments we use the trained model checkpoints and supervised scores from Locatello et al. (2018) to evaluate $\beta$-VAE, CCI-VAE, FactorVAE, TC-VAE, DIP-VAE-I and DIP-VAE-II on two benchmark datasets: dSprites (Matthey et al., 2017) and 3D Shapes (Burgess & Kim, 2018) (see Sec. A.3 for details). Each model is trained with $H = 6$ different hyperparameter settings (detailed in Sec. A.4.1 in Supplementary Material), with $S = 50$ seeds per setting, and $P = 50$ pairwise comparisons.

**UDR correlates well with the supervised metrics.** To validate UDR, we calculate Spearman's correlation between its model ranking and that produced by four existing supervised disentanglement metrics found to be the most meaningful in the large scale comparison study by Locatello et al. (2018): the original $\beta$-VAE metric (Higgins et al., 2017a), FactorVAE metric (Kim & Mnih, 2018), Mutual Information Gap (MIG) (Chen et al., 2018) and DCI Disentanglement (Eastwood & Williams, 2018) (see Sec. A.6 in Supplementary Material for metric details). The average correlation for UDR

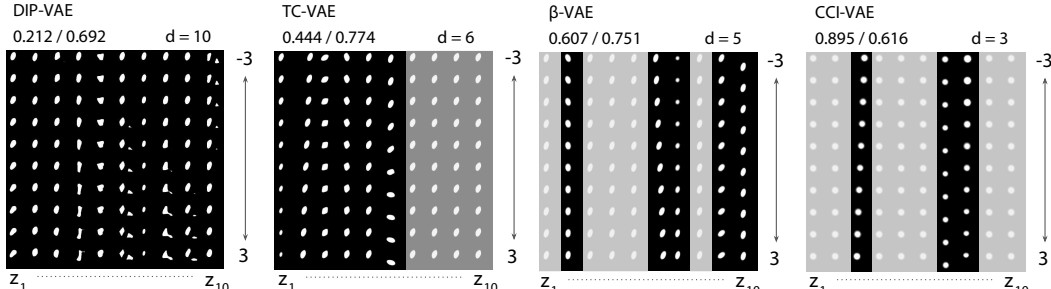

Figure 3: Latent traversals of the top ranked trained DIP-VAE-I, TC-VAE, CCI-VAE and $\beta$-VAE according to the UDR method. At the top of each plot the two presented scores are UDR/FactorVAE metric. Note that the FactorVAE metric scores visually entangled models very highly. $d$ is the number of informative latents. The uninformative latents are greyed out.

is $0.54 \pm 0.06$ and for UDR-A2A is $0.60 \pm 0.11$. This is comparable to the average Spearman's correlation between the model rankings produced by the different supervised metrics: $0.67 \pm 0.2$. The variance in rankings produced by the different metrics is explained by the fact that the metrics capture different aspects of disentangling (see Sec. A.2 in Supplementary Materials for a discussion of how UDR relates to other representation comparison methods). Tbl. 1 provides a breakdown of correlation scores between MIG and the different versions of UDR for different model classes and datasets. It is clear that the different versions of UDR perform similarly to each other, and this holds across datasets and model classes. Note that unlike the supervised metrics, UDR does not assume a "canonical" disentangled representation. Instead, it allows any one of the many equivalent possible ground truth generative processes to become the "canonical" one for each particular dataset and model class, as per the theoretical results by Rolinek et al. (2019) summarised in Sec. 3.1.

**UDR is useful for hyperparameter selection.** Fig. 2 compares the scores produced by UDR and the four supervised metrics for 3600 trained models, split over six model classes, two datasets and six hyperparameter settings. We consider the median score profiles across the six hyperparameter settings to evaluate whether a particular setting is better than others. It can be seen that UDR broadly agrees with the supervised metrics on which hyperparameters are more promising for disentangling. This holds across datasets and model classes. Hence, UDR may be useful for evaluating model fitness for disentangled representation learning as part of an evolutionary algorithm or Bayesian hyperparameter tuning.

**UDR is useful for model selection.** Fig. 2 can also be used to examine whether a particular trained model has learnt a good disentangled representation. We see that some models reach high UDR scores. For example, more models score highly as the value of the $\beta$ hyperparameter is increased in the $\beta$-VAE model class. This is in line with the previously reported results (Higgins et al., 2017a). Note that the 0th hyperparameter setting in this case corresponds to $\beta = 1$, which is equivalent to the standard VAE objective (Kingma & Welling, 2014; Rezende et al., 2014). As expected, these models score low in terms of disentangling. We also see that for some model classes (e.g. DIP-VAE-I, DIP-VAE-II and FactorVAE on dSprites) no trained model scores highly according to UDR. This suggests that none of the hyperparameter choices explored were good for this particular dataset, and that no instance of the model class learnt to disentangle well. To test this, we use latent traversals to qualitatively evaluate the level of disentanglement achieved by the models, ranked by their UDR scores. This is a common technique to qualitatively evaluate the level of disentanglement on simple visual datasets where no ground truth attribute labels are available. Such traversals involve changing the value of one latent dimension at a time and evaluating its effect on the resulting reconstructions to understand whether the latent has learnt to represent anything semantically meaningful. Fig. 3 demonstrates that the qualitative disentanglement quality is reflected well in the UDR scores. The figure also highlights that the supervised metric scores can sometimes be overoptimistic. For example, compare TC-VAE and $\beta$-VAE traversals in Fig. 3. These are scored similarly by the supervised metric (0.774 and 0.751) but differently by UDR (0.444 and 0.607). Qualitative evaluation of the traversals clearly shows that $\beta$-VAE learnt a more disentangled representation than TC-VAE, which is captured by UDR but not by the supervised metric. Fig. 9 in Supplementary Material provides more examples. We also evaluated how well UDR ranks models trained on more complex datasets, CelebA and ImageNet, and found that it performs well (see Sec. A.9 in Supplementary Materials).

| SAMPLE # ($P$) | 5 | 10 | 15 | 20 | 25 | 30 | 35 | 40 | 45 |
|---|---|---|---|---|---|---|---|---|---|
| CORRELATION | 0.51±0.07 | 0.57±0.03 | 0.57±0.05 | 0.6±0.03 | 0.59±0.03 | 0.61±0.02 | 0.61±0.02 | 0.61±0.01 | 0.61±0.01 |

Table 2: Rank correlations of the UDR score with the $\beta$-VAE metric on the dSprites dataset for a $\beta$-VAE hyperparameter search as the number of pairwise comparisons $P$ per model were changed.

**UDR works well even with five pairwise comparisons.** We test the effect of the number of pairwise comparisons $P$ on the variance and accuracy of the UDR scores. Tbl. 2 reports the changes in the rank correlation with the $\beta$-VAE metric on the dSprites dataset as $P$ is varied between 5 and 45. We see that the correlation between the UDR and the $\beta$-VAE metric becomes higher and the variance decreases as the number of seeds is increased. However, even with $P=5$ the correlation is reasonable.

**UDR generalises to a dataset with no attribute labels.** We investigate whether UDR can be useful for selecting well disentangled models trained on the 3D Cars (Reed et al., 2014) dataset with poorly labelled attributes, which makes it a bad fit for supervised disentanglement metrics. Fig. 1 shows that a highly ranked model according to UDR appears disentangled, while a poorly ranked one appears entangled. Fig. 10 in Supplementary Material provides more examples of high and low scoring models according to the UDR method.

**UDR predicts final task performance.** We developed UDR to help practitioners use disentangled representations to better solve subsequent tasks. Hence, we evaluate whether the model ranking produced by UDR correlates with task performance on two different domains: the fairness on a classification task introduced by Locatello et al. (2018), and data efficiency on a clustering task for a model-based reinforcement learning agent introduced by Watters et al. (2019) (see Sec. A.8 in Supplementary Materials for more details). We found that UDR had an average of 0.8 Spearman correlation with the fairness scores, which is higher than the average of 0.72 correlation between fairness and supervised scores reported by Locatello et al. (2018). We also found that UDR scores had 0.56 Spearman correlation with data efficiency of the COBRA agent. The difference between the best and the worst models according to UDR amounted to around 66% reduction in the number of steps to 90% success rate on the task.

## 6 CONCLUSION

We have introduced UDR, the first method for unsupervised model selection for variational disentangled representation learning. We have validated our approach on 5400 models covering all six state of the art VAE-based unsupervised disentangled representation learning model classes. We compared UDR to four existing supervised disentanglement metrics both quantitatively and qualitatively, and demonstrated that our approach works reliably well across three different datasets, often ranking models more accurately than the supervised alternatives. Moreover, UDR avoids one of the big shortcomings of the supervised disentangling metrics – the arbitrary choice of a "canonical" disentangled factorisation, instead allowing any of the equally valid disentangled generative processes to be accepted. Finally, we also demonstrated that UDR is useful for predicting final task performance using two different domains. Hence, we hope that UDR can be a step towards unlocking the power of unsupervised disentangled representation learning to real-life applications.

## ACKNOWLEDGEMENTS

We thank Olivier Bachem and Francesco Locatello for helping us re-use their code and model checkpoints, and Neil Rabinowitz, Avraham Ruderman and Tatjana Chavdarova for useful feedback.

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

# A SUPPLEMENTARY MATERIAL

## A.1 USEFUL PROPERTIES OF DISENTANGLED REPRESENTATIONS

Disentangled representations are particularly useful because they re-represent the information contained in the data in a way that enables semantically meaningful *compositionality*. For example, having discovered that the data is generated using two factors, colour and shape, such a model would be able to support meaningful reasoning about fictitious objects, like pink elephants, despite having never seen one during training (Higgins et al., 2017b; 2018b). This opens up opportunities for counterfactual reasoning, more robust and interpretable inference and model-based planning (Higgins et al., 2018a; Suter et al., 2018). Furthermore, such a representation would support more data efficient learning for subsequent tasks, like a classification objective for differentiating elephants from cats. This could be achieved by ignoring the nuisance variables irrelevant for the task, e.g. the colour variations, by simply masking out the disentangled subspaces that learnt to represent such nuisances, while only paying attention to the task-relevant subspaces, e.g. the units that learnt to represent shape (Cohen & Welling, 2016; Gens & Domingos, 2014; Soatto, 2010). Hence, the semantically meaningful compositional nature of disentangled representations is perhaps the most sought after aspect of disentangling, due to its strong implications for generalisation, data efficiency and interpretability (Schmidhuber, 1992; Bengio et al., 2013; Higgins et al., 2018a).

## A.2 ASPECTS OF DISENTANGLEMENT MEASURED BY DIFFERENT METRICS

Methods for evaluating and comparing representations have been proposed in the past. The most similar approaches to ours are the DCI Disentanglement score from Eastwood & Williams (2018) and the axis alignment comparison of representations in trained classifiers proposed in Li et al. (2016). The former is not directly applicable for unsupervised disentangled model selection, since it requires access to the ground truth attribute labels. Even when adapted to compare two latent representations, our preliminary experiments suggested that the entropy based aggregation proposed in Eastwood & Williams (2018) is inferior to our aggregation in Eq. 2 when used in the UDR setup. The approach by Li et al. (2016) shares the similarity matrix calculation step with us, however they never go beyond that quantitatively, opting for qualitative evaluations of model representations instead. Hence, their approach is not directly applicable to quantitative unsupervised disentangled model ranking.

Other related approaches worth mentioning are the Canonical Correlation Analysis (CCA) and its modifications (Hardoon et al., 2004; Raghu et al., 2017; Morcos et al., 2018). These approaches, however, tend to be invariant to invertible affine transformations and therefore to the axis alignment of individual neurons, which makes them unsuitable for evaluating disentangling quality. Finally, Representation Similarity Matrix (RSM) (Kriegeskorte et al., 2008) is a commonly used method in Neuroscience for comparing the representations of different systems to the same set of stimuli. This technique, however, is not applicable for measuring disentangling, because it ignores dimension-wise response properties.

When talking about disentangled representations, three properties are generally considered: *modularity*, *compactness* and *explicitness*[2] (Ridgeway & Mozer, 2018). *Modularity* measures whether each latent dimension encodes only one data generative factor, *compactness* measures whether each data generative factor is encoded by a single latent dimension, and *explicitness* measures whether all the information about the data generative factors can be decoded from the latent representation. We believe that *modularity* is the key aspect of disentangling, since it measures whether the representation is compositional, which gives disentangled representations the majority of their beneficial properties (see Sec. A.1 in Supplementary Materials for more details). *Compactness*, on the other hand, may not always be desirable. For example, according to a recent definition of disentangled representations (Higgins et al., 2018a), it is theoretically impossible to represent 3D rotation in a single dimension (see also Ridgeway & Mozer (2018)). Finally, while *explicitness* is clearly desirable for preserving information about the data that may be useful for subsequent tasks, in practice models often fail to discover and represent the full set of the data generative factors due to restrictions on both the observed data distribution and the model capacity (Mathieu et al., 2019). Hence, we suggest noting the explicitness of a representation, but not necessarily punishing its disentanglement ranking if it is not fully explicit. Instead, we suggest that the practitioner should have the choice to select the most disentangled model given a particular number of discovered generative factors. Hence, in the rest of the paper we will use the term "disentanglement" to refer to the compositional property of a representation related to the modularity measure. Tbl. 3 provides a summary of how the different metrics considered in the paper compare in terms of modularity, compactness and explicitness.

## A.3 DATASET DETAILS

**dSprites** A commonly used unit test for evaluating disentangling is the dSprites dataset (Matthey et al., 2017). This dataset consists of images of a single binary sprite pasted on a blank background and can be fully described by five generative factors: shape (3 values), position x (32 values), position y (32 values), size (6 values) and

---

[2]Similar properties have also been referred to as *disentanglement*, *completeness* and *informativeness* respectively in the independent yet concurrent paper by Eastwood & Williams (2018).

Table 3: Disentangled model selection metrics comparison. M - modularity, C - compactness, E - explicitness (Ridgeway & Mozer, 2018)

| METRIC | M | C | E |
|---|---|---|---|
| $\beta$-VAE | $\checkmark$ | $\times$ | $\checkmark$ |
| FACTORVAE | $\checkmark$ | $\checkmark$ | $\checkmark$ |
| MIG | $\checkmark$ | $\checkmark$ | $\checkmark$ |
| DCI DISENTANGLEMENT | $\checkmark$ | $\times$ | $\times$ |
| UDR | $\checkmark$ | $\times$ | $\times$ |

rotation (40 values). All the generative factors are sampled from a uniform distribution. Rotation is sampled from the full 360 degree range. The generative process for this dataset is fully deterministic, resulting in 737,280 total images produced from the Cartesian product of the generative factors.

**3D Shapes**   A more complex dataset for evaluating disentangling is the 3D Shapes dataset (Burgess & Kim, 2018). This dataset consists of images of a single 3D object in a room and is fully specified by six generative factors: floor colour (10 values), wall colour (10 values), object colour (10 values), size (8 values), shape (4 values) and rotation (15 values). All the generative factors are sampled from a uniform distribution. Colours are sampled from the circular hue space. Rotation is sampled from the [-30, 30] degree angle range.

**3D Cars**   This dataset was adapted from Reed et al. (2014). The full data generative process for this dataset is unknown. The labelled factors include 199 car models and 24 rotations sampled from the full 360 degree out of plane rotation range. An example of an unlabelled generative factor is the colour of the car – this varies across the dataset.

### A.4   UNSUPERVISED DISENTANGLED REPRESENTATION LEARNING MODELS

As mentioned in Sec. 3, current state of the art approaches to unsupervised disentangled representation learning are based on the VAE (Kingma & Welling, 2014; Rezende et al., 2014) objective presented in Eq. 1. These approaches decompose the objective in Eq. 1 into various terms and change their relative weighting to exploit the trade-off between the capacity of the latent information bottleneck with independent sources of noise, and the quality of the resulting reconstruction in order to learn a disentangled representation. The first such modification was introduced by Higgins et al. (2017a) in their $\beta$-VAE framework:

$$\mathbb{E}_{p(\boldsymbol{x})}[\, \mathbb{E}_{q_\phi(\boldsymbol{z}|\boldsymbol{x})}[\log p_\theta(\boldsymbol{x}|\boldsymbol{z})] - \beta \, KL(q_\phi(\boldsymbol{z}|\boldsymbol{x}) \,||\, p(\boldsymbol{z})) \,] \tag{4}$$

In order to achieve disentangling in $\beta$-VAE, the KL term in Eq. 4 is typically up-weighted by setting $\beta > 1$. This implicitly reduces the latent bottleneck capacity and, through the interaction with the reconstruction term, encourages the generative factors $c_k$ with different reconstruction profiles to be encoded by different independent noisy channels $z_l$ in the latent bottleneck. Building on the $\beta$-VAE ideas, CCI-VAE (Burgess et al., 2017) suggested slowly increasing the bottleneck capacity during training, thus improving the final disentanglement and reconstruction quality:

$$\mathbb{E}_{p(\boldsymbol{x})}[\, \mathbb{E}_{q_\phi(\boldsymbol{z}|\boldsymbol{x})}[\log p_\theta(\boldsymbol{x}|\boldsymbol{z})] - \gamma \, |KL(q_\phi(\boldsymbol{z}|\boldsymbol{x}) \,||\, p(\boldsymbol{z})) - C| \,] \tag{5}$$

Later approaches (Kim & Mnih, 2018; Chen et al., 2018; Kumar et al., 2017) showed that the KL term in Eq. 1 can be further decomposed according to:

$$\mathbb{E}_{p(\boldsymbol{x})}[\, KL(q_\phi(\boldsymbol{z}|\boldsymbol{x}) \,||\, p(\boldsymbol{z})) \,] = I(\boldsymbol{x};\boldsymbol{z}) + KL(q_\phi(\boldsymbol{z}) \,||\, p(\boldsymbol{z})) \tag{6}$$

Hence, penalising the full KL term as in Eqs. 4-5 is not optimal, since it unnecessarily penalises the mutual information between the latents and the data. To remove this undesirable side effect, different authors suggested instead adding more targeted penalised terms to the VAE objective function. These include different implementations of the total correlation penalty (FactorVAE by Kim & Mnih (2018) and TC-VAE by Chen et al. (2018)):

$$\mathcal{L}_{VAE} - \gamma \, KL(q_\phi(\boldsymbol{z}) \,||\, \prod_{j=1}^{M} q_\phi(z_j)) \tag{7}$$

and different implementations of the penalty that pushes the marginal posterior towards a factorised prior (DIP-VAE by Kumar et al. (2017)):

$$\mathcal{L}_{VAE} - \gamma \, KL(q_\phi(\boldsymbol{z}) \,||\, p(\boldsymbol{z})) \tag{8}$$

### A.4.1   MODEL IMPLEMENTATION DETAILS

We re-used the trained checkpoints from Locatello et al. (2018), hence we recommend the readers to check the original paper for model implementation details. Briefly, the following architecture and optimiser were used.

Table 4: Encoder and Decoder Implementation details shared for all models

| Encoder | Decoder |
|---|---|
| Input: $64 \times 64 \times$ number of channels | Input: $\mathbb{R}^{10}$ |
| $4 \times 4$ conv, 32 ReLU, stride 2 | FC, 256 ReLU |
| $4 \times 4$ conv, 32 ReLU, stride 2 | FC, $4 \times 4 \times 64$ ReLU |
| $4 \times 4$ conv, 64 ReLU, stride 2 | FC, $4 \times 4$ upconv, 64 ReLU, stride 2 |
| $4 \times 4$ conv, 64 ReLU, stride 2 | FC, $4 \times 4$ upconv, 32 ReLU, stride 2 |
| FC 256, F2 $2 \times 10$ | $4 \times 4$ upconv, 32 ReLU, stride 2 |
| | $4 \times 4$ upconv, number of channels, stride 2 |

Table 5: Hyperparameters used for each model architecture

| Model | Parameters | Values |
|---|---|---|
| $\beta$-VAE | $\beta$ | [1, 2, 4, 6, 8, 16] |
| CCI-VAE | $c_{\max}$ | [5, 10, 25, 50, 75, 100] |
| | iteration threshold | 100000 |
| | $\gamma$ | 1000 |
| FactorVAE | $\gamma$ | [10, 20, 30, 40, 50, 100] |
| DIP-VAE-I | $\lambda_{od}$ | [1, 2, 5, 10, 20, 50] |
| | $\lambda_d$ | $10\lambda_{od}$ |
| DIP-VAE-II | $\lambda_{od}$ | [1, 2, 5, 10, 20, 50] |
| | $\lambda_d$ | $\lambda_{od}$ |
| TC-VAE | $\beta$ | [1, 2, 4, 6, 8, 10] |

(a) Common hyperparameters across all models

| Parameter | Values |
|---|---|
| Batch Size | 64 |
| Latent space dimension | 10 |
| Optimizer | Adam |
| Adam: beta1 | 0.9 |
| Adam: beta2 | 0.999 |
| Adam: epsilon | 1e-8 |
| Adam: learning rate | 0.0001 |
| Decoder type | Bernoulli |

(b) FactorVAE discriminator architecture

| Discriminator |
|---|
| FC, 1000 leaky ReLU |
| FC, 1000 leaky ReLU |
| FC, 1000 leaky ReLU |
| FC, 1000 leaky ReLU |
| FC, 1000 leaky ReLU |
| FC, 1000 leaky ReLU |
| FC, 2 |

(c) FactorVAE discriminator parameters

| Parameter | Values |
|---|---|
| Batch size | 64 |
| Optimizer | Adam |
| Adam: beta1 | 0.5 |
| Adam: beta2 | 0.9 |
| Adam: epsilon | 1e-8 |
| Adam: learning rate | 0.0001 |

Table 6: Miscellaneous model details

For consistency, all the models were trained using the same architecture, optimiser, and hyperparameters. All of the methods use a deep neural network to encode and decode the latent embedding and the parameters of the latent factors are predicted using a Gaussian encoder whose architecture is specified in Table 4. All of the models predict a latent vector with 10 factors. Each model was also trained with 6 different levels of regularisation strength specified in Table 5. The ranges of the hyperparameters used for the various levels of regularisation were specified to show a diversity of different performance on different datasets without relying on pre-existing intuition on good hyperparameters, however ranges were based on hyperparameters that were used previously in literature. For each of the model classes outlined above, we tried 6 hyperparameter values with 50 seeds each.

$\beta$-**VAE** The $\beta$-VAE (Higgins et al., 2017a) model is similar to the vanilla VAE model but with an additional hyperparameter $\beta$ to modify the strength of the KL regulariser.

$$\mathbb{E}_{p(\boldsymbol{x})}\big[\,\mathbb{E}_{q_\phi(\boldsymbol{z}|\boldsymbol{x})}[\log p_\theta(\boldsymbol{x}|\boldsymbol{z}] - \beta\,KL(q_\phi(\boldsymbol{z}|\boldsymbol{x})\,||\,p(\boldsymbol{z}))\,\big] \tag{9}$$

where a $\beta$ value of 1 corresponds to the vanilla VAE model. Increasing $\beta$ enforces a stronger prior on the latent distribution and encourages the representation to be independent.

**CCI-VAE**    The CCI-VAE model (Burgess et al., 2017) is a variant of the $\beta$-VAE where the KL divergence is encouraged to match a controlled value $C$ which is increased gradually throughout training. This yields us the objective function for CCI-VAE.

$$\mathbb{E}_{p(\boldsymbol{x})}\big[\,\mathbb{E}_{q_\phi(\boldsymbol{z}|\boldsymbol{x})}[\log p_\theta(\boldsymbol{x}|\boldsymbol{z}] - \beta\,|KL(q_\phi(\boldsymbol{z}|\boldsymbol{x})\,||\,p(\boldsymbol{z})) - C|\,\big] \tag{10}$$

**FactorVAE**    FactorVAE (Kim & Mnih, 2018) specifically penalises the dependencies between the latent dimensions such that the "Total Correlation" term is targeted yielding a modified version of the $\beta$-VAE objective.

$$\mathbb{E}_{p(\boldsymbol{x})}\big[\,\mathbb{E}_{q_\phi(\boldsymbol{z}|\boldsymbol{x})}[\log p_\theta(\boldsymbol{x}|\boldsymbol{z}] - KL(q_\phi(\boldsymbol{z}|\boldsymbol{x})\,||\,p(\boldsymbol{z}))\,\big]$$
$$-\beta KL(q(\boldsymbol{z})||\prod_j q(\boldsymbol{z}_j)) \tag{11}$$

The "Total Correlation" term is intractable in this case so for FactorVAE, samples are used from both $q(\boldsymbol{z}|\boldsymbol{x})$ and $q(\boldsymbol{z})$ as well as the density-ratio trick to compute an estimate of the "Total Correlation" term. FactorVAE uses an additional discriminator network to approximate the density ratio in the KL divergence. The implementation details for the discriminator network and its hyperparameters can be found in Table 5(b) and 5(c).

**TC-VAE**    The TC-VAE model (Chen et al., 2018) which independently from FactorVAE has a similar objective KL regulariser which contains a "Total Correlation" term. In the case of TC-VAE the "Total Correlation" term is estimated using a biased Monte-Carlo estimate.

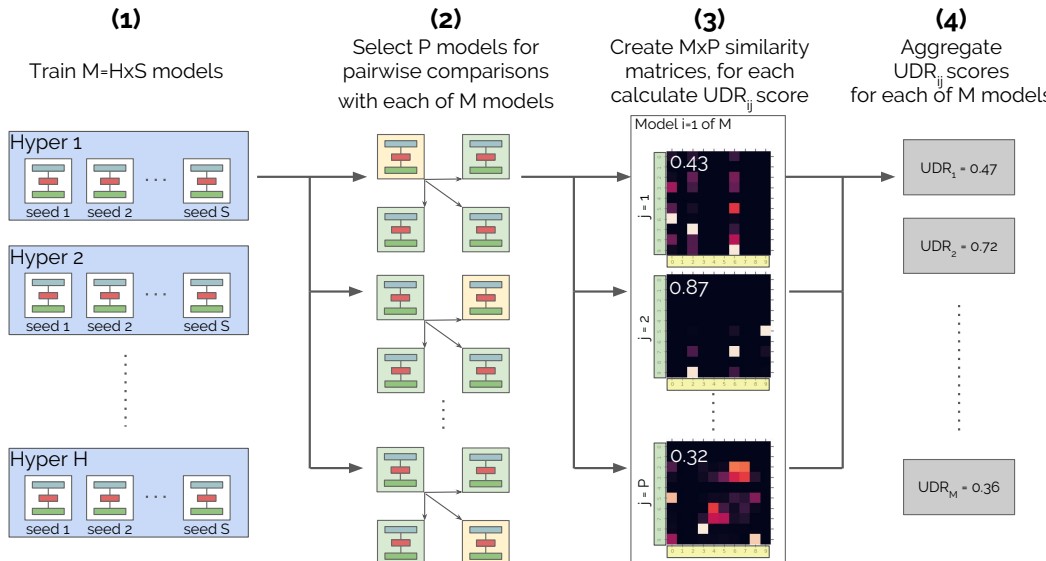

Figure 4: Schematic illustration of the UDR method. See details in text.

**DIP-VAE**    The DIP-VAE model also adds regularisation to the aggregated posterior but instead an additional loss term is added to encourage it to match the factorised prior. Since the KL divergence is intractable, other measures of divergence are used instead. $Cov_{p(\boldsymbol{x})}[\mu_\phi(\boldsymbol{x})]$ can be used, yielding the DIP-VAE-I objective

$$\mathbb{E}_{p(\boldsymbol{x})}\big[\,\mathbb{E}_{q_\phi(\boldsymbol{z}|\boldsymbol{x})}[\log p_\theta(\boldsymbol{x}|\boldsymbol{z}] - KL(q_\phi(\boldsymbol{z}|\boldsymbol{x})\,||\,p(\boldsymbol{z}))\,\big]$$
$$-\lambda_{od}\sum_{i\neq j}[Cov_{p(x)}[\mu_\phi(\boldsymbol{x})]]_{ij}^2$$
$$-\lambda_d\sum_i([Cov_{p(\boldsymbol{x})}[\mu_\phi(\boldsymbol{x})]]_{ii}-1)^2 \tag{12}$$

or $Cov_{q_\phi}[\boldsymbol{z}]$ is used instead yielding the DIP-VAE-II objective.

$$
\begin{aligned}
\mathbb{E}_{p(\boldsymbol{x})}\big[\,\mathbb{E}_{q_\phi(\boldsymbol{z}|\boldsymbol{x})}[\log p_\theta(\boldsymbol{x}|\boldsymbol{z})] - KL(q_\phi(\boldsymbol{z}|\boldsymbol{x})\,\|\,p(\boldsymbol{z}))\,\big] & \\
-\lambda_{od}\sum_{i\neq j}[Cov_{q_\phi}[\boldsymbol{z}]]_{ij}^2 & \\
-\lambda_d\sum_i([Cov_{q_\phi}[\boldsymbol{z}]]_{ii}-1)^2 &
\end{aligned}
\tag{13}
$$

## A.5 UDR IMPLEMENTATION DETAILS

**Similarity matrix**  To compute the similarity matrix $R_{ij}$ we follow the approach of Li et al. (2016) and Morcos et al. (2018). For a given dataset $X=\{\boldsymbol{x}_1,\boldsymbol{x}_2,....,\boldsymbol{x}_N\}$ and a neuron $a\in\{1,...,L\}$ of model $i$ (denoted as $z_{i,a}$), we define $\boldsymbol{z}_{i,a}$ to be the vector of mean inferred posteriors $q_i(\boldsymbol{z}_i|\boldsymbol{x}_i)$ across the full dataset: $\boldsymbol{z}_{i,a}=(z_{i,a}(\boldsymbol{x}_1),...,z_{i,a}(\boldsymbol{x}_N))\in\mathbb{R}^N$. Note that this is different from the often considered notion of a "latent representation vector". Here $\boldsymbol{z}_{i,a}$ is a response of a single latent dimension over the entire dataset, not an entire latent response for a single input. We then calculate the similarity between each two of such vectors $\boldsymbol{z}_{i,a}$ and $\boldsymbol{z}_{j,b}$ using either Lasso regression or Spearman's correlation.

**Lasso regression (UDR$_L$)**  We trained $L$ lasso regressors to predict each of the latent responses $z_{i,a}$ from $\boldsymbol{z}_j$ using the dataset of latent encodings $Z_{i,a}=\{(\boldsymbol{z}_{j,1},z_{i,a,1}),...,(\boldsymbol{z}_{j,N},z_{i,a,N})\}$. Each row in $R_{ij}(a)$ is then filled in using the weights of the trained Lasso regressor for $z_{i,a}$. The lasso regressors were trained using the default Scikit-learn multi-task lasso objective $\min_w \frac{1}{2n_{samples}}||XW-Y||_{Fro}^2+\lambda||W||_{21}$ where $Fro$ is the Frobenius norm: $||A||_{Fro}=\sqrt{\sum_{ij}a_{ij}^2}$ and the $l_1l_2$ loss is computed as $||A||_{21}=\sum_i\sqrt{\sum_j a_{ij}^2}$. $\lambda$ is chosen using cross validation and the lasso is trained until convergence until either 1000 iterations have been run or our updates are below a tolerance of 0.0001. Lasso regressors were trained on a dataset of 10000 latents from each model and training was performed using coordinate descent over the entire dataset. $R_{nm}$ is then computed by extracting the weights in the trained lasso regressor and computing their absolute value (Eastwood & Williams, 2018). It is important that the representations are normalised per-latent such that the relative importances computed per latent are scaled to reflect their contribution to the output. Normalising our latents also ensures that the weights that are computed roughly lie in the interval $[-1,1]$.

**Spearman's based similarity matrix (UDR$_S$)**  We calculate each entry in the similarity matrix according to $R_{ij}(a,b)=\text{Corr}(\boldsymbol{z}_{i,a},\boldsymbol{z}_{j,b})$, where Corr stands for Spearman's correlation. We use Spearman's correlation to measure the similarity between $\boldsymbol{z}_{i,a}$ and $\boldsymbol{z}_{j,b}$, because we do not want to necessarily assume a linear relationship between the two latent encodings, since the geometry of the representational space is not crucial for measuring whether a representation is disentangled (see Sec. 2), but we do hope to find a monotonic dependence between them. Spearman correlation coefficients were computed by extracting 1000 samples from each model and computing the Spearman correlation over all the samples on a per-latent basis.

**All-to-all calculations**  To make all-to-all comparisons, we picked 10 random seeds per hyperparameter setting and limited all the calculations to those models. Hence we made the maximum of (60 choose 2) pairwise model comparisons when calculating UDR-A2A.

**Informative latent thresholding**  Uninformative latents typically have KL$\ll$0.01 while informative latents have KL$\gg$0.01, so KL$=$0.01 threshold in Eq. 3 is somewhat arbitrarily chosen to pick out the informative latents $z$.

**Sample reduction experiments**  We randomly sampled without replacement 20 different sets of $P$ models for pairwise comparison from the original set of 50 models with the same hyperparameter setting for UDR or 60 models with different seeds and hyperparameters for UDR-A2A.

## A.6 SUPERVISED METRIC IMPLEMENTATION DETAILS

**Original $\beta$-VAE metric.**  First proposed in Higgins et al. (2017a), this metric suggests sampling two batches of observations $\boldsymbol{x}$ where in both batches the same single data generative factor is fixed to a particular value, while the other factors are sampled randomly from the underlying distribution. These two batches are encoded into the corresponding latent representations $q_\phi(\boldsymbol{z}|\boldsymbol{x})$ and the pairwise differences between the corresponding mean latent values from the two batches are taken. Disentanglement is measured as the ability of a linear classifier to predict the index of the data generative factor that was fixed when generating $\boldsymbol{x}$.

We compute the $\beta$-VAE score by first randomly picking a single factor of variation and fixing the value of that factor to a randomly sampled value. We then generate two batches of 64 where all the other factors are sampled

randomly and take the mean of the differences between the latent mean responses in the two batches to generate a training point. This process is repeated 10000 times to generate a training set by using the fixed factor of variation as the label. We then train a logistic regression on the data using Scikit-learn and report the evaluation accuracy on a test set of 5000 as the disentanglement score.

**FactorVAE metric.** Kim & Mnih (2018) proposed a modification on the $\beta$-VAE metric which made the classifier non-parametric (majority vote based on the index of the latent dimension with the least variance after the pairwise difference step). This made the FactorVAE metric more robust, since the classifier did not need to be optimised. Furthermore, the FactorVAE metric is more accurate than the $\beta$-VAE one, since the $\beta$-VAE metric often over-estimates the level of disentanglement by reporting 100% disentanglement even when only $K-1$ factors were disentangled.

The Factor VAE score is computed similarly to the $\beta$-VAE metric but with a few modifications. First we draw a set of 10000 random samples from the dataset and we estimate the variance of the mean latent responses in the model. Latents with a variance of less than 0.05 are discarded. Then batches of 64 samples are generated by a random set of generative factors with a single fixed generative factor. The variances of all the latent responses over the 64 samples are computed and divided by the latent variance computed in the first step. The variances are averaged to generate a single training point using the fixed factor of variation as the label. 10000 such training points are generated as the training set. A majority vote classifier is trained to pick out the fixed generative factor and the evaluation accuracy is computed on test set of 5000 and reported as the disentanglement score.

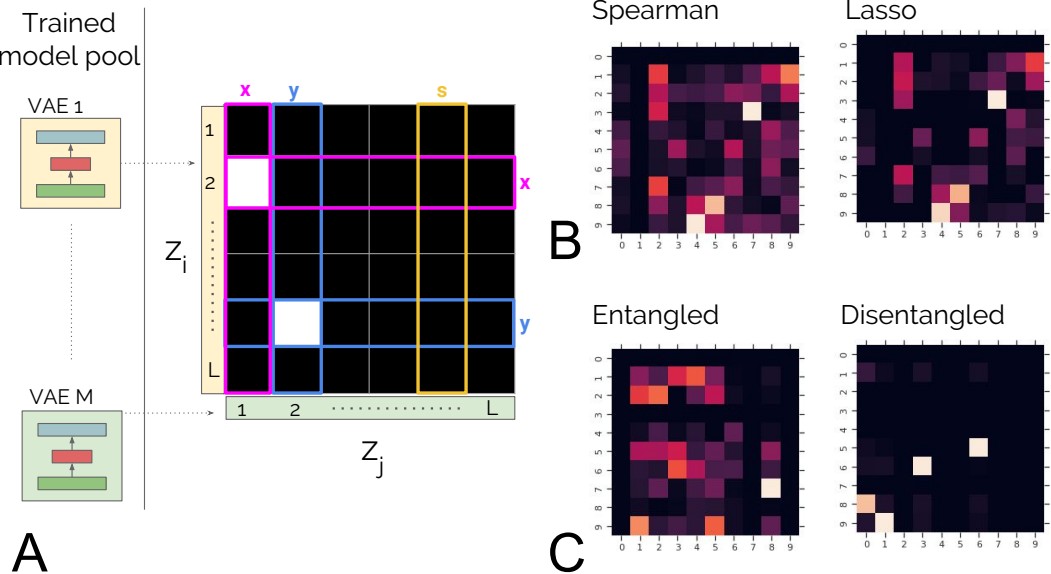

Figure 5: **A:** Schematic illustration of the pairwise model comparison. Two trained models $i$ and $j$ are sampled for pairwise comparison. Both models learnt a perfectly disentangled representation, learning to represent two (positions x/y) and three (positions x/y, and size) generative factors respectively. Similarity matrix $R_{ij}$: white – high similarity between latent dimensions, black – low. **B:** Similarity matrix $R_{ij}$ for the same pair of models, calculated using either Spearman correlation or Lasso regression. The latter is often cleaner. **C:** Examples of Lasso similarity matrices of an entangled vs a disentangled model.

**Mutual Information Gap (MIG).** The MIG metric proposed in Chen et al. (2018) proposes estimating the mutual information (MI) between each data generative factor and each latent dimension. For each factor, they consider two latent dimensions with the highest MI scores. It is assumed that in a disentangled representation only one latent dimension will have high MI with a single data generative factor, and hence the difference between these two MI scores will be large. Hence, the MIG score is calculated as the average normalised difference between such pairs of MI scores per each data generative factor. Chen et al. (2018) suggest that the MIG score is more general and unbiased than the $\beta$-VAE and FactorVAE metrics.

We compute the Mutual Information Gap by taking the discretising the mean representation of 10000 samples into 20 bins. The disentanglement score is then derived by computing, per generative factor, the difference between the top two latents with the greatest mutual information with the generative factor and taking the mean.

Table 7: Rank correlations between each of the scores produced by the four versions of UDR and four supervised metrics. The scores are averaged over three model classes, two datasets and four supervised metrics. See Supplementary Material for details.

| UDR | LASSO | SPEARMAN | SUPERVISED |
|---|---|---|---|
| HYPER | $0.54\pm0.06$ | $0.53\pm0.07$ | $0.67\pm0.2$ |
| ALL-TO-ALL | $0.60\pm0.11$ | $0.59\pm0.10$ | |

$$\frac{1}{K}\sum_{k=1}^{K}\frac{1}{H_{v_k}}\Big(I(z_j^{(k)}) - \max_{j\neq j_k}I(z_j,v_k)\Big) \tag{14}$$

where K is the number of generative factors, from which $v_k$ is a single generative factor $z_j$ is the mean representation and $j^{(}k) = \mathrm{argmax}_j I_n(z_j;v_k)$ is the latent representation with the greatest mutual information with the generative factor. $H_{v_k}$ is the computed entropy of the generative factor.

**DCI Disentanglement.** This is the disentanglement part of the three-part metric proposed by Eastwood & Williams (2018). The DCI disentanglement metric is somewhat similar to our unsupervised metric, whereby the authors train a random forest classifier to predict the ground truth factors from the corresponding latent encodings $q(\boldsymbol{z}|\boldsymbol{x})$. They then use the resulting $M\times N$ matrix of feature importance weights to calculate the difference between the entropy of the probability that a latent dimension is important for predicting a particular ground truth factor weighted by the relative importance of each dimension.

The DCI disentanglement metric is an implementation of the disentanglement metric as described in Eastwood & Williams (2018) using a gradient boosted tree. It was computed by first extracting the relative importance of each latent mean representation as a predictor for each generative factor by training a gradient boosted tree using the default Scikit-learn model on 10000 training and 1000 test points and extracting the importance weights. The weights are summarised into an importance matrix $R_{ij}$ with the number of rows equal to the number of generative factors and columns equal to the number of latents. The disentanglement score for each column is computed as $D_i = (1 - H_K(P_i))$ where $H_K(P_i) = -\sum_{k=0}^{K-1}P_{ik}log_K P_{ik}$ denotes the entropy. $P_{ik} = R_{ij}/\sum_{k=0}^{K-1}$ is the probability of the latent factor $i$ in being important for predicting factor $k$. The weighted mean of the scores for the column is computed using the relative predictive importance of each column as the weight $D = \sum_i p_i * D_i$ where $p_i = \sum_j R_{ij}/\sum_{ij}R_{ij}$.

## A.7 ADDITIONAL RESULTS

We evaluated four UDR versions, which differed in terms of whether Spearman- and Lasso-based similarity matrices $R_{ij}$ were used (subscripts S and L respectively), and whether the models for pairwise similarity comparison are picked from the pool of different seeds trained with the same hyperparameters or from the pool of all models (the latter indicated by the A2A suffix). The A2A correlations in Tbl. 7 are on average slightly higher, however these scores are more computationally expensive to compute due to the higher number of total pairwise similarity calculations. For that reason, the scores presented in the table are calculated using only 20% of all the trained models. Hence, the results presented in the main text of the paper are computed using the UDR$_L$ score, which allowed us to evaluate all 5400 models and performed slightly better than the UDR$_S$ score. Figs. 6-8 provide more details on the performance of the different UDR versions.

To qualitatively validate that the UDR method is ranking models well, we look into more detail into the $\beta$-VAE model ranking when evaluated with the DCI disentanglement metric on the dSprites dataset. This scenario resulted in the worst disagreement between UDR and the supervised metric as shown in Fig. 6. We consider the UDR$_L$ version of our method, since it appears to give the best trade off between overall correlations with the supervised metrics and hyperparameter selection accuracy. Fig. 9 demonstrates that the poor correlation between UDR$_L$ and DCI Disentanglement is due to the supervised metric. Models ranked highly by UDR$_L$ but poorly by DCI Disentanglement appear to be qualitatively disentangled through visual inspection of latent traversals. Conversely, models scored highly by DCI Disentanglement but poorly by UDR$_L$ appear entangled.

## A.8 UDR CORRELATION WITH FINAL TASK PERFORMANCE

To illustrate the usefulness of UDR to select disentangled models, we ran two experiments. We computed the UDR correlation with fairness scores and with data efficiency on a model-based RL task.

**Fairness scores.** Fig. 11 (left) demonstrates that UDR correlates well with the classification fairness scores introduced by Locatello et al. (2019). We adopted a similar setup described in Locatello et al. (2019) to compute fairness, using a gradient booting classifier over 10000 labelled examples. The fairness score was computed by taking

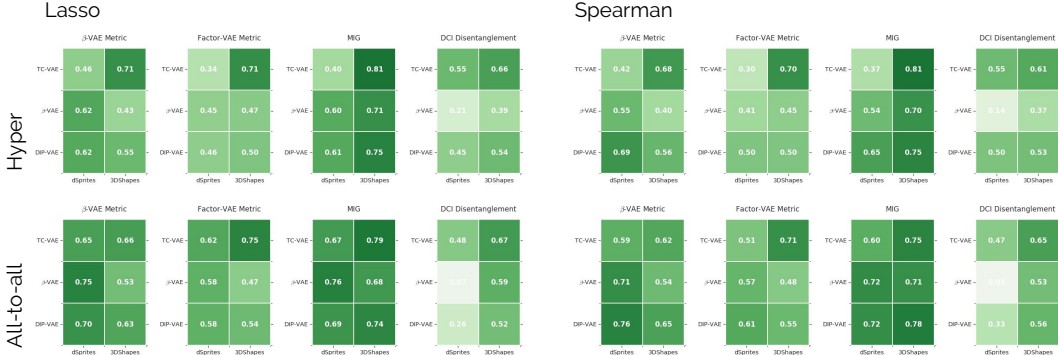

Figure 6: Rank correlation between different versions of UDR with different supervised metrics across two datasets and three model classes. We see that the UDR$_L$ approaches slightly outperform the UDR$_S$ ones.

the mean of the fairness scores across all targets and all sensitive variables where the fairness scores are computed by measuring the total variation after intervening on the sensitive variable. The fairness scores were compared against the Lasso regression version of UDR where models were paired only within the same hyperparameters.

**Model-based RL data efficiency.** We reproduced the results from the COBRA agent (Watters et al., 2019), to observe if UDR would correlate with the final tasks performance when using VAEs as state representations. More precisely, we will look at the training data efficiency, reported as the number of steps needed to achieve 90% performance on the Clustering tasks (see Watters et al. (2019) for details), while using differently disentangled models.

The agent is provided with a pre-trained MONet (Burgess et al., 2019), an exploration policy and a transition model and has to learn a good reward predictor for the task in a dense reward setting. It uses Model Predictive Control in order to plan and solve the task, where sprites have to be clustered by color (e.g. two blue sprites and two red sprites). In COBRA, the authors use a MONet with disentangled representation by using a high $\beta = 1$.

When pre-training MONet, we used $\beta \in \{0.01, 0.1, 1\}$ in order to introduce entanglement in the representations without compromising reconstruction accuracy and pre-trained 10 seeds for each value of $\beta$. We use 5 random initialisations of the reward predictor for each possible MONet model, and train them to perform the clustering task as explained in Watters et al. (2019). We report the number of steps to reach 90% success, averaged across the initialisations. The UDR score is computed by feeding images with a single sprite to obtain an associated unique representation and proceeding as described in the main text.

As can be seen in Figure 11 (right), we find that the UDR scores correlate with this final data efficiency (linear regression shown, Spearman correlation $\rho = 0.56$). This indicates that one could leverage the UDR score as a metric to select representations for further tasks. In this analysis we used the version of UDR that uses Spearman correlations and within-hyperparameter model comparisons.

## A.9 EVALUATING UDR ON MORE COMPLEX DATASETS

We evaluated whether UDR is useful for model selection on more complex datasets. In particular, we chose CelebA and ImageNet. While disentangling VAEs have been shown to perform well on CelebA in the past (e.g. Higgins et al. (2018b)), ImageNet is notoriously too complex for even vanilla VAEs to model. However, we still wanted to verify whether the coarse representations of VAEs on ImageNet could be disentangled, and if so, whether UDR would be useful for model selection. To this end, we ran a hyperparameter sweep for the $\beta$-VAE and ranked its representations using UDR. Fig. 12 shows that UDR scores are clearly different for the different values of the $\beta$ hyperparameter. It is also clear that the models were able to learn about CelebA and produce reasonable reconstructions, but on ImageNet even the vanilla VAEs struggled to represent anything but the coarsest information. Figs. 13-14 plot latent traversals for three randomly chosen models with high (>0.6) and low (<0.3) UDR scores. The latents are sorted by their informativeness, as approximated by their batch-averaged per dimension KL with the prior as per Eq. 3. It is clear that for both datasets those models that are ranked high by the UDR have both more interpretable and more similar representations than those models that are ranked low.

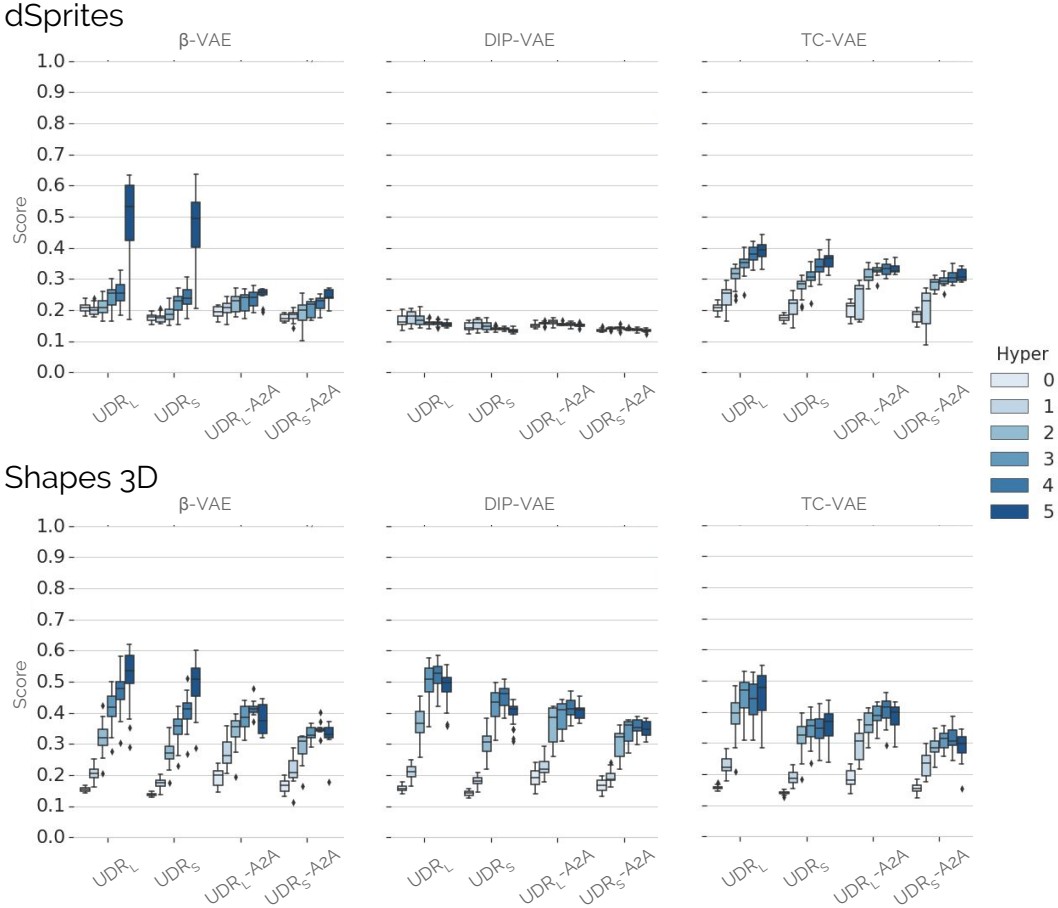

Figure 7: The range of scores for each hyperparameter setting for the dSprites and 3D Shapes datasets for various models and metrics. We see that the different versions of the UDR method broadly agree with each other.

### A.10 QUALITATIVE EVALUATION OF MODEL REPRESENTATIONS RANKED BY UDR SCORES

In this section we attempt to qualitatively verify our assumption that "for a particular dataset and a VAE-based unsupervised disentangled representation learning model class, disentangled representations are all alike, while every entangled representation is entangled in its own way, to rephrase Tolstoy". The theoretical justification of the proposed UDR hinges on the work by Rolinek et al. (2019). However, that work only empirically evaluated their analysis on the $\beta$-VAE model class. Even though we have reasons to believe that their theoretical results would also hold for the other disentangling VAEs evaluated in this paper, in this section we empirically evaluate whether this is true.

First, we check if all model classes operate in the so called "polarised regime", which was highlighted by Rolinek et al. (2019) as being important for pushing VAEs towards disentanglement. It is known that even vanilla VAEs (Kingma & Welling, 2014; Rezende et al., 2014) enter the "polarised regime", which is often cited as one of their shortcomings (e.g. see Rezende & Viola (2018)). All of the disentangling VAEs considered in this paper augment the original ELBO objective with extra terms. None of these extra terms penalise entering the "polarised regime", apart from that of DIP-VAE-I. We tested empirically whether different model classes entered the "polarised regime" during our hyperparameter sweeps. We did this by counting the number of latents that were "switched off" in each of the 5400 models considered in our paper by using Eq. 3. We found that all models apart from DIP-VAE-I entered the polarised regime during the hyperparameter sweep, having on average 2.95/10 latents "switched off" (with a standard deviation of 1.97).

Second, we check if the models scored highly by the UDR do indeed have similar representations, and models that are scored low have dissimilar representations. We do this qualitatively by plotting latent traversals for three randomly chosen models within each of the six disentangling VAE model classes considered in this paper.

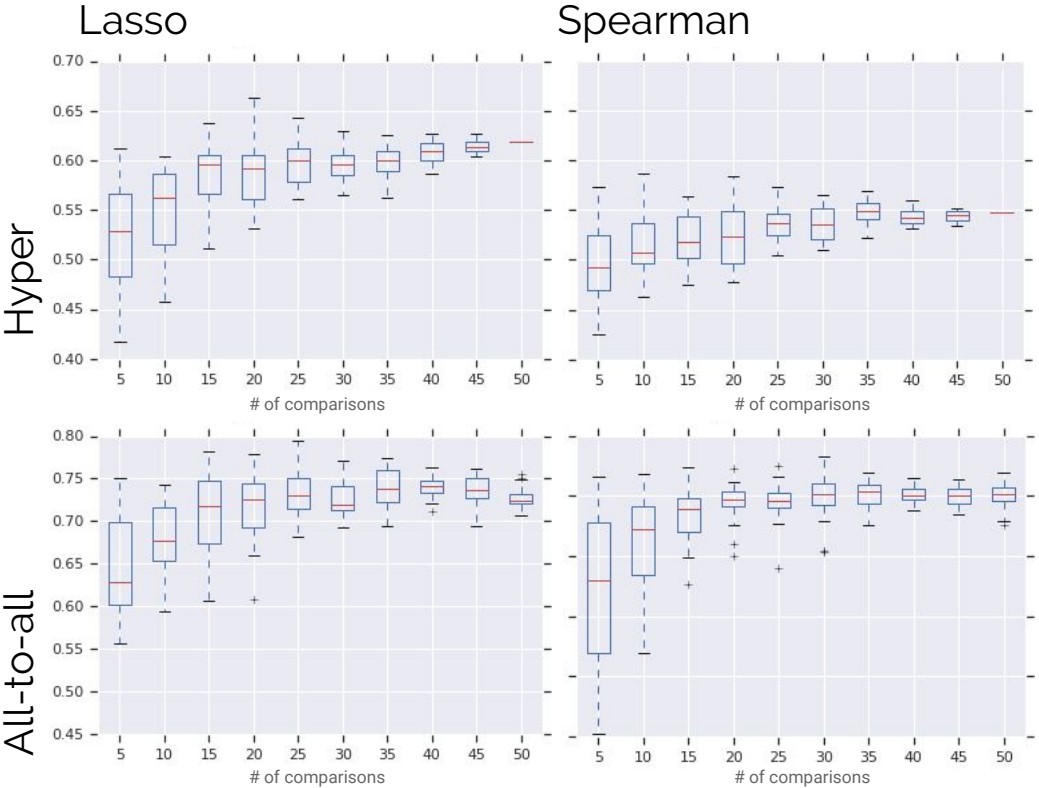

Figure 8: Rank correlations of the different versions of the UDR score with the $\beta$-VAE metric on the dSprites dataset for a $\beta$-VAE hyperparameter search as the number of pairwise comparisons per model were changed. Higher number of comparisons leads to more accurate and more stable rankings, however these are still decent even with 5 pairwise comparisons per model.

We groups these plots by UDR scores into three bands: high (UDR>0.4), medium (0.3<UDR<0.4) and low (UDR<0.3). Fig. 15 shows latent traversals for all model classes that were able to achieve the high range of UDR scores (note that some model classes were not able to achieve high UDR values with the hyperparameter settings evaluated in this paper). We present the latent traversals for all ten latents per model without sorting them in any particular way. We also colour code the latents by their semantic meaning (if the meaning is apparent from the latent traversal). Fig. 15 shows that the representations learnt by the highly ranked models all appear to be very similar (up to subsetting, sign inverse and permutation). Note that the models also include many latents that are "switched off". Fig. 16 shows latent traversals for two model classes that did not achieve high UDR scores. We see that these models now have fewer semantically meaningful latent dimensions, and fewer latents that are uninformative or "switched off". Finally, Fig. 17 shows latent traversals for all model classes that had models which scored low on UDR. We see that many of these models do not have any uninformative latents, and their representations are hard to interpret. Furthermore, it is hard to find similarity between the representations learnt by the different models. Together, Figs. 15-17 empirically verify that our assumption holds for the model classes considered in this paper. However, we recommend that any practitioner using UDR on new disentangling model classes developed in the future first verify that the assumptions of the UDR hold for those models. We suggest training the new models on a number of toy but well studied datasets (like dSprites or Shapes3D) and checking if the ranks produced by UDR correlate with those produced by the supervised metrics. Furthermore, we suggest a qualitative evaluation of the traversals plots for the high and low scoring models.

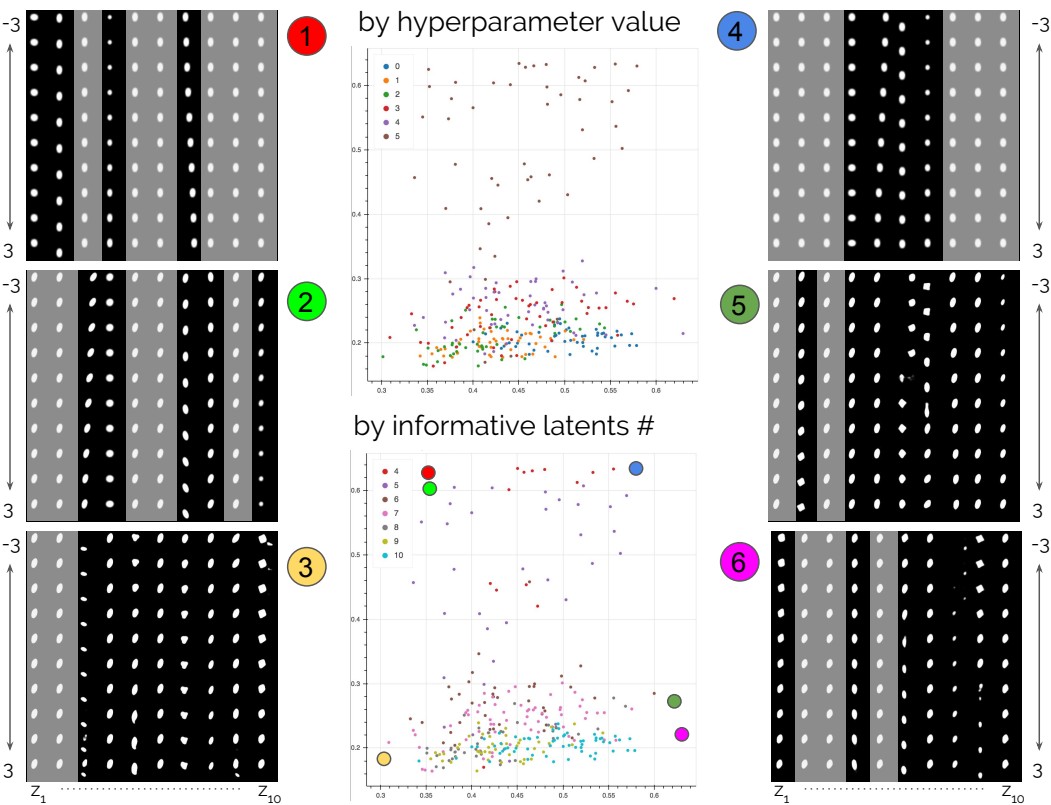

Figure 9: Example latent traversals of some of the best and worst ranked $\beta$-VAE models using the UDR$_L$ (ordinate) and DCI Disentanglement (abscissa) metrics, coloured either by hyperparameter value (top) or final informative latent number (bottom). Uninformative units are greyed out. The models ranked highly by UDR$_L$ do appear to be well disentangled, despite being ranked poorly by DCI Disentanglement (1, 2, 4). On the other hand, models ranked well by DCI Disentanglement but poorly by UDR$_L$ look quite entangled (5, 6). Finally, models ranked poorly by both metrics do appear entangled (3).

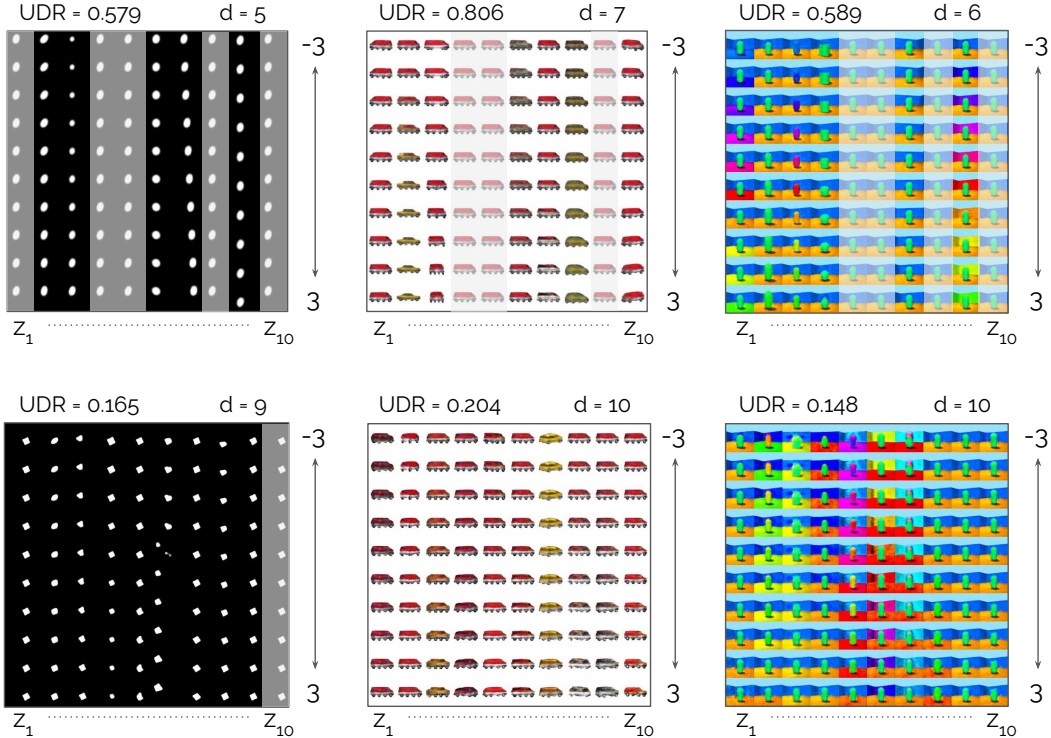

Figure 10: Example latent traversals of some of the best and worst ranked $\beta$-VAE models using the UDR$_L$ scores. Uninformative latents are greyed out.

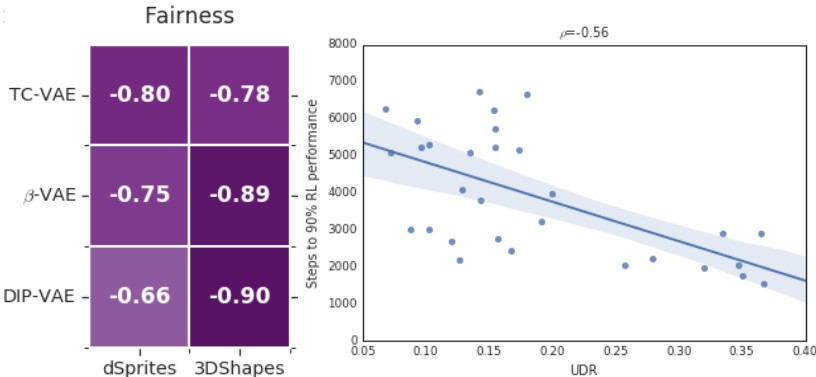

Figure 11: **Left**: Spearman correlation between UDR scores and classification fairness scores introduced by Locatello et al. (2019) across sixty models trained per each one of the three different model classes (rows) and over two datasets (columns). **Right**: Spearman correlation between UDR scores and data efficiency for learning a clustering task by the COBRA agent introduced by Watters et al. (2019). Lower step number is better.

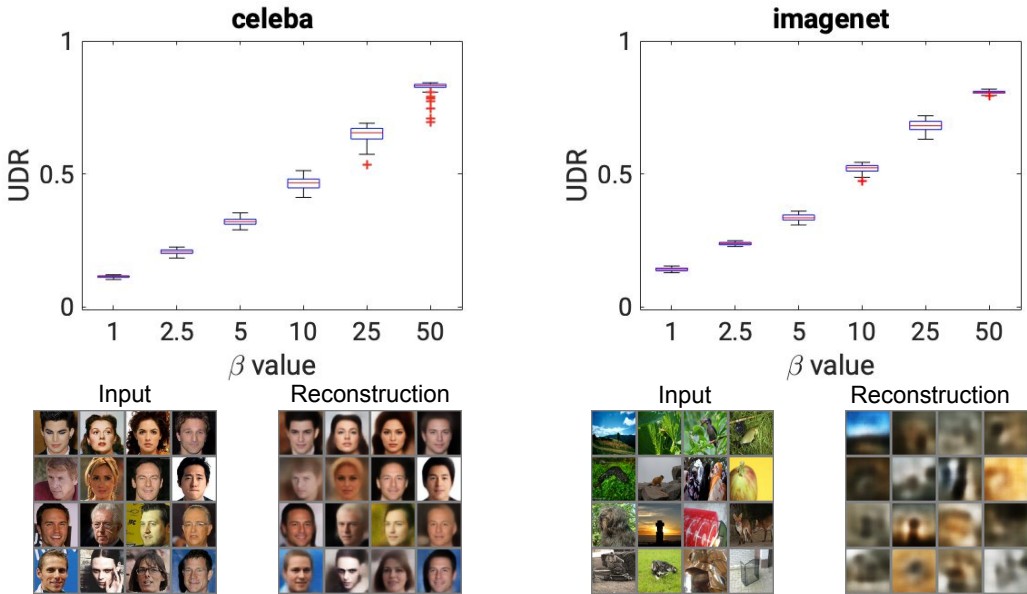

Figure 12: Distribution of UDR$_S$ scores (P=50) for 300 $\beta$-VAE models trained with 6 settings of the $\beta$ hyperparameter and 50 seeds on CelebA and ImageNet datasets. The reconstructions shown are for the vanilla VAE ($\beta = 1$). ImageNet is a complex dataset that VAEs struggle to model well.

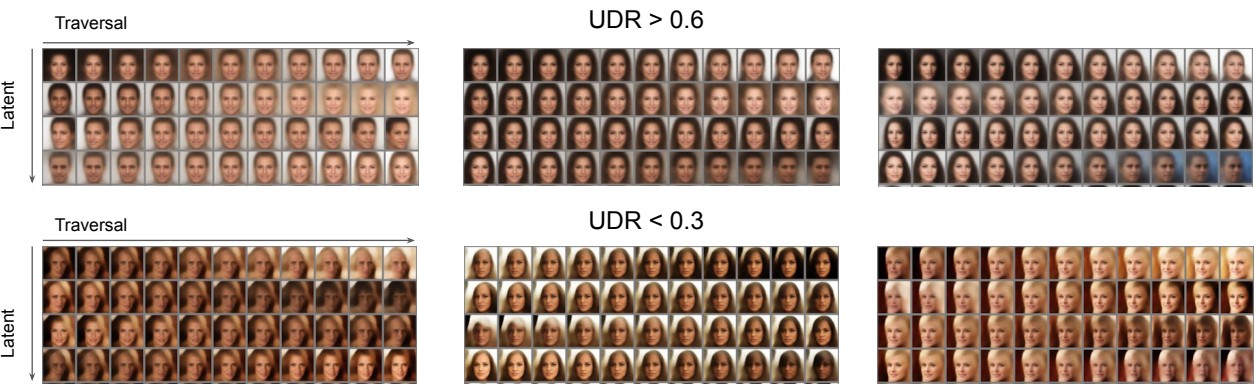

Figure 13: Latent traversals for the four most informative latents ordered by their KL from the prior for three different $\beta$-VAE models that ranked high or low according to UDR. Those models that were ranked high have learnt representations that are both interpretable and very similar across models. Those models that were ranked low have learnt representations that are harder to interpret and they do not appear similar to each other across models.

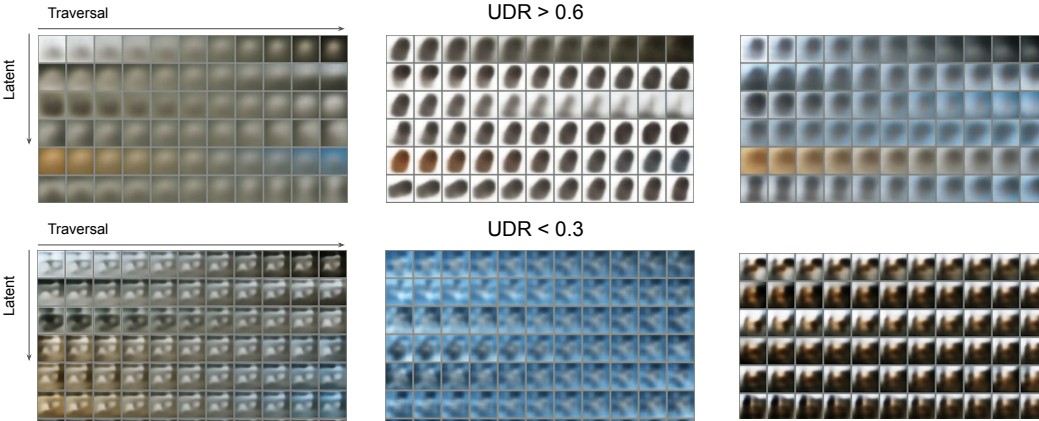

Figure 14: Latent traversals for the six most informative latents ordered by their KL from the prior for three different $\beta$-VAE models that ranked high or low according to UDR. Despite the fact that none of the $\beta$-VAE or VAE models were able to learn to reconstruct this dataset well, those models that were ranked high by the UDR still managed to learn representations that are more interpretable and more similar across models. This is unlike the representations of those models that were ranked low by the UDR.

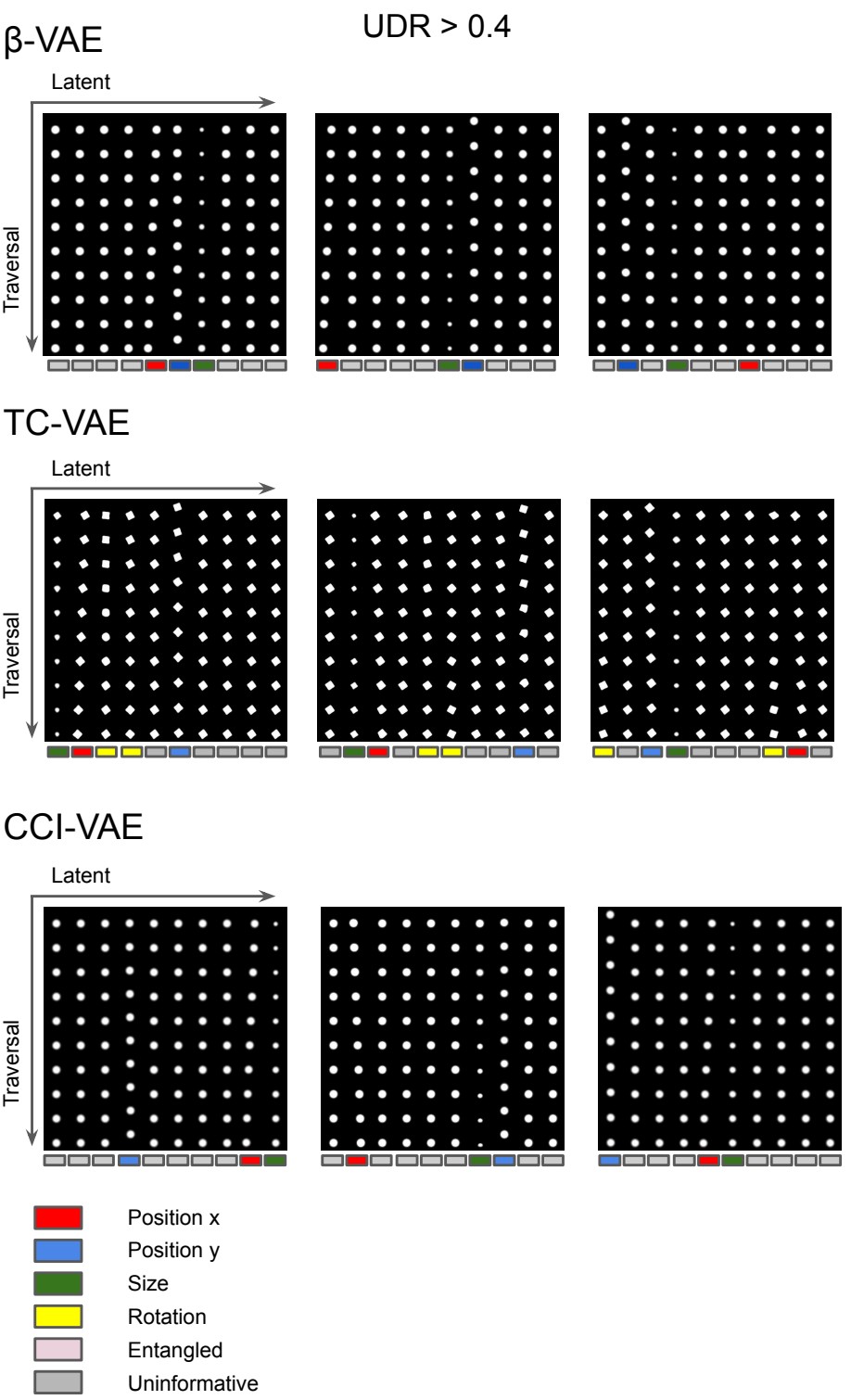

Figure 15: Latent traversals for all ten latent dimensions presented in no particular ordering for three different models per model class. These models were ranked highly by the UDR. It can be seen that they learnt interpretable and similar representations up to permutation, sign inverse and subsetting. We included all model classes that achieved UDR scores in the range specified (UDR > 0.4).

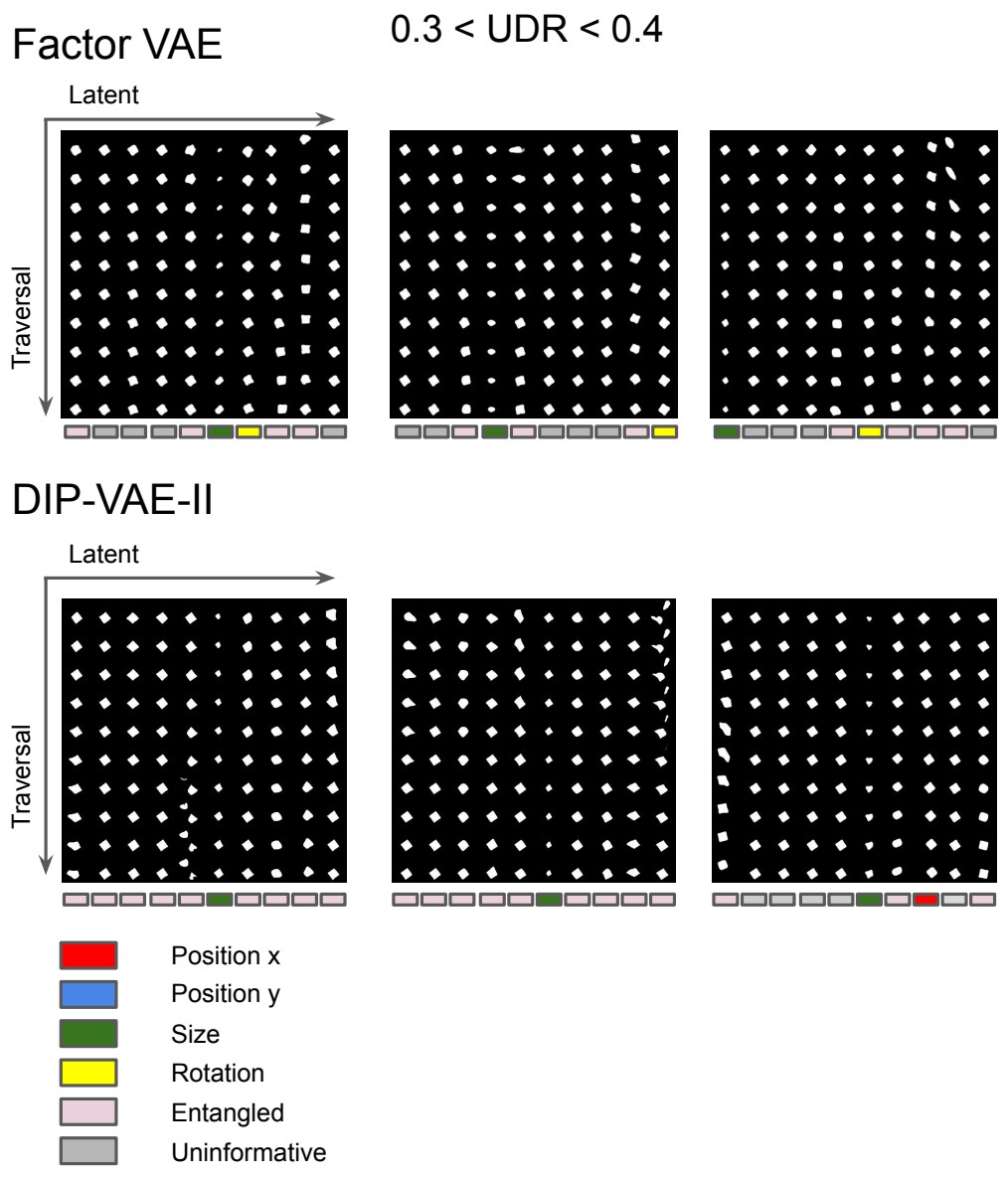

Figure 16: Latent traversals for all ten latent dimensions presented in no particular ordering for three different models per model class. These models received medium UDR scores. It can be seen that they learnt less interpretable and less similar representations than the models shown in Fig. 15. None of the models in these model classes scored higher than the range specified (0.3 < UDR < 0.4).

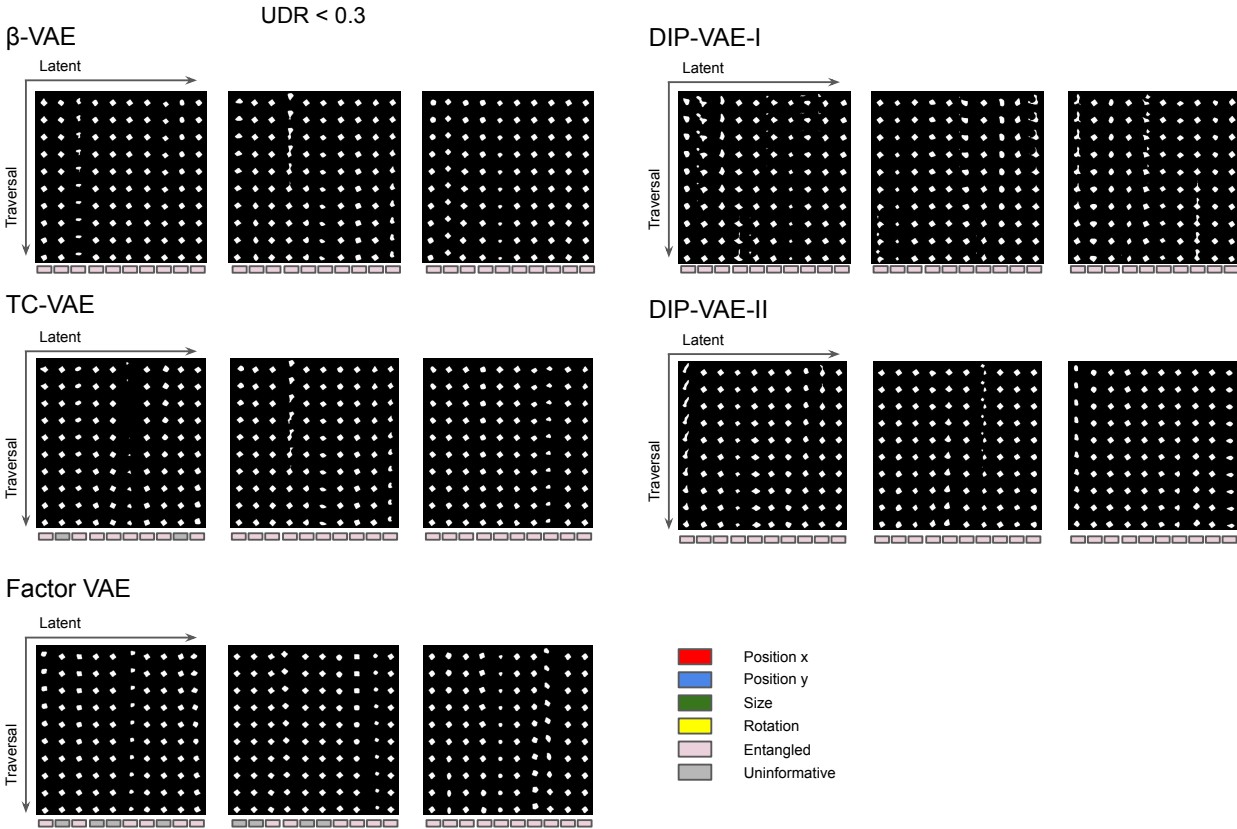

Figure 17: Latent traversals for all ten latent dimensions presented in no particular ordering for three different models per model class. These models received low UDR scores. It can be seen that their representations are hard to interpret and they look quite different from each other. We included all model classes that achieved UDR scores in the range specified (UDR < 0.3).

