# OpenReview forum: "Unsupervised Model Selection for Variational Disentangled Representation Learning"
_ICLR.cc/2020/Conference — Accept (Poster)_

### Official Review · AnonReviewer2 · 2019-10-23
**Official Blind Review #2**

**Rating:** 6

**Review:**

The paper proposes a metric for unsupervised model (and hyperparameter) selection for VAE-based models. The essential basis for the metric is to rank the models based on how much disentanglement they provide. This method relies on a key observation from this paper [A] viz., disentangled representations by any VAE-based model are likely to be similar (upto permutation and sign).

I am inclined to accept the paper for the following reasons:
1. The proposed approach is clear and easy enough to understand and well motivated
2. The paper has clearly outlined the assumptions and limitations of their work
3. The reported result show that models ranked by disentanglement correlate well with the supervised metrics for the various VAE models.
4. This metric is unsupervised and thus can utilize far more data than the supervised metric methods and can be useful even when the dataset has no labels.
5. The supplementary material also shows that the metric correlates well with the task performance.

[A] Variational Autoencoders Pursue PCA Directions (by Accident), CVPR 2019

---

Update:

Thanks for the thoughtful rebuttal by the authors to all the reviewers' feedback.

Based on the several discussions by the other reviewers and the discussion that happened, I am inclined to lower my scores to a weak accept.

**Experience Assessment:**

I do not know much about this area.

**Review Assessment: Checking Correctness Of Derivations And Theory:**

N/A

**Review Assessment: Checking Correctness Of Experiments:**

I assessed the sensibility of the experiments.

**Review Assessment: Thoroughness In Paper Reading:**

I read the paper at least twice and used my best judgement in assessing the paper.

---

> ### Author Response · Authors · 2019-11-08
> **Response to Review2**
>
> Dear Reviewer, thank you for your feedback.

---

### Official Review · AnonReviewer1 · 2019-10-23
**Official Blind Review #1**

**Rating:** 6

**Review:**

This paper proposes a criterion called Unsupervised Disentanglement Ranking (UDR) score. The score is computed based on the assumption that good disentangled representations are alike, while the representations can be entangled in multiple possible ways.  The UDR score can be used for unsupervised hyperparameter tuning and model selection for variational disentangled method.

The problem this paper focuses on is essential because we usually apply unsupervised disentangled methods to analyze the data when the labels are unavailable. However, existing metrics for hyperparameter tuning and model selection requires ground-truth labels. This paper allows measuring model performance without supervision, making the hyperparameter tuning and model selection possible in practice.

It looks like some parts of this paper need rewriting. In the abstract, it is not mentioned at all what is the proposed approach. Most paragraphs in the introduction section review the related work and background but do not introduce what assumption and strategy the proposed method adopted.

It looks like the proposed UDR is theoretically supported by Rolinek et al. (2019). However, the proof given by Rolinek et al. (2019) is for $\beta$-VAE, where the regularization can be turned into the constraint on KL divergence. I do not think the "polarised regime" holds for other disentangled model, for example, TCVAE, where a biased estimation of total correlation is introduced in the objective function. Therefore, I am not convinced that I should trust the results of the UDR, which combines multiple disentangled models.

The computational process of UDR is heuristic and somewhat arbitrary. There is no theoretical guarantee that UDR should be a useful disentanglement metric.  Although the UDR is supported by some experiments, I am not convinced that it is trustworthy for more complex real-world datasets.

Equation (3) looks problematic. Note that it is possible to train a Bidirectional Generative Adversarial Network (BiGAN) that can generate complex images based on a uniform distribution (Donahue et al., 2016). The encoder of the BiGAN can be considered as the inverse of the generator, which maps images back to the uniform distribution. This suggests that under the encoder-decoder framework, it is possible that latent variables can be informative even the posterior distribution matches the prior distribution. Although VAEs are trained using a different strategy, I do not see why the posterior needs to diverge from the prior distribution for informative latent representations. The encoder might simply be the inverse of the decoder under a certain scenario.

In summary, this paper focuses on solving an important problem. However, the proposed method is not well supported by theorems as it seems. The paper also appears to contain minor technical issues. Therefore, I am inclined to reject this paper.

References
Donahue, Jeff, Philipp Krähenbühl, and Trevor Darrell. "Adversarial feature learning." arXiv preprint arXiv:1605.09782 (2016).

**Experience Assessment:**

I have read many papers in this area.

**Review Assessment: Checking Correctness Of Derivations And Theory:**

I carefully checked the derivations and theory.

**Review Assessment: Checking Correctness Of Experiments:**

I assessed the sensibility of the experiments.

**Review Assessment: Thoroughness In Paper Reading:**

I read the paper thoroughly.

---

> ### Author Response · Authors · 2019-11-08
> **Response to Reviewer1**
>
> Dear Reviewer,
>
> Thank you for your thoughtful comments. In your feedback you have brought up three major points: 1) you have expressed doubt whether the “polarised regime” holds for other disentangling methods other than \beta-VAE; 2) you were wondering what underlies the thinking behind the choice of our computational process; and 3) you were wondering about our choice of using per latent KL divergence to identify which latents are informative. We will address these points below.
>
>
> 1) You have expressed doubt whether the “polarised regime” holds for other disentangling methods other than \beta-VAE, which was used as the example model in Rolinek et al. First, even the vanilla VAEs (Kingma and Welling, 2014) are known to enter the “polarised regime”, which is often cited as one of their shortcomings (e.g. see Rezende and Viola, 2018). All of the disentangling VAEs considered in the paper, including TC-VAE, augment the original ELBO objective with extra terms. Hence, all of them still contain the \beta KL term of the ELBO with \beta => 1. This means that in theory all of them inherit the property of the original VAEs of operating in the “polarised regime”. We have tested this empirically by counting the number of latents that are “switched off” in each of the 5400 models considered in our paper. We found that all models apart from DIP-VAE-I entered the polarised regime, having on average 2.95/10 latents “switched off” (with a standard deviation of 1.97). Note that DIP-VAE-I is the only model with an objective that explicitly penalises “switching off” latent dimensions, which means that it is less suitable for disentangled representation learning in the common regime where the number of generative factors is smaller than the number of latents (as discussed in the original paper by Kumar et al, 2018). Despite DIP-VAE-I never entering the polarised regime, the results reported in our paper (e.g. in Fig. 2 or Fig. 6 in Supplementary Materials) suggest that our proposed UDR still performs well and correlates highly with the supervised metrics for this model class.
>
> 2) The computational process proposed in our paper is motivated by the theoretical results presented in Rolinek et al. Unfortunately there is no computationally feasible way to calculate directly whether the SVD decomposition J=UΣV of the Jacobian J of the decoder results in a V, which is a signed permutation matrix. Our approach uses a simple process to approximate this in a computationally feasible way. We have applied our proposed method to a number of datasets commonly used in the literature and have demonstrated that it performs well across 5400 models. Please let us know if you have a particular suggestion for a more complex dataset that you would like to test our metric on.
>
> 3) In terms of Equation 3, the GAN objective in the BiGAN paper implicitly minimises the KL divergence between the prior p(z) and the marginalised posterior q(z). However, in Eq.3 we measure the KL divergence between the prior and the conditional posterior q(z|x). Hence, the two are not directly comparable. Eq. 3 is a way to quantify which latent dimensions are used by the network that has entered the “polarised regime”. Rolinek et al define a model to be in the “polarised regime” if its latents can be split into two disjoint sets of “used” and “unused” dimensions. “Used” dimensions are defined as those which have inferred \sigma^2 << 1, and the “unused” dimensions are defined as those which have \mu^2 << 1 and \sigma^2 \approx 1 (see Sec 3.2, Definition 1 in Rolinek et al). Note that the latter would result in a small KL from a unit Gaussian prior as per Eq. 3 in our paper, thus justifying our choice to find the “used” and “unused” latents.
>
> Finally, we have modified the abstract and introduction to mention our proposed method as per your suggestion.

---

> > ### Comment · AnonReviewer1 · 2019-11-11
> > **Response to authors**
> >
> > Thanks for your response.
> >
> > 1) I understand that the "polarised regime" holds for vanilla VAE and $\beta$-VAE, as introduced by Rolinek et al. However, this does not imply that after augmenting the ELBO with an arbitrary extra term, the "polarised regime" still holds in theory. As you mentioned above, DIP-VAE-I is one of such examples that the "polarised regime" does not hold. It is also not clear to me whether it holds for other methods in theory.
> >
> > If the proposed method is based on the "polarised regime" assumption, then I think you might need to either prove that the "polarised regime" holds for other methods, or you might need to test whether it holds for each seed and hyperparameter with experiments. I think the simplest fix might be adding one more step in UDR computation, which tests whether the "polarised regime" holds for each model and only include the results where the "polarised regime" holds (This implies that we should exclude DIP-VAE-I).
> >
> > It looks to me that the proposed method does not explicitly make use of the "polarised regime" assumption. If the "polarised regime" assumption is not necessary, then some rewritings might be necessary. The current version seems to suggest that the "polarised regime" is the key assumption of the proposed method.
> >
> > The above discussion also makes me curious about how UDR performs if we include $\beta$-VAE only. We are sure that the "polarised regime" holds in this case.
> >
> > 2) Thanks for your clarification.
> >
> > It would be more convincing to me if the proposed method performs well on the ImageNet dataset, or any text and genetic datasets. I think these are the datasets better simulate real-world scenarios where unsupervised analysis is necessary. The included datasets are all synthetic datasets. It is not surprising that different disentanglement methods give a consensus on these simpler datasets.
> >
> > 3) Thanks for your clarification.

---

> > > ### Author Response · Authors · 2019-11-12
> > > **Made changes to the paper and started extra experiments**
> > >
> > > Dear Reviewer, thank you for getting back to us.
> > >
> > > 1) We agree that there are no guarantees that the other disentangling models necessarily enter the “polarised regime”. However, the reason why we were quite confident that this would happen in practice for the other methods (apart from DIP-VAE-I) is that the extra disentanglement terms that these approaches add to the ELBO do not act against the VAE’s tendency to "switch off" latent dimensions. DIP-VAE-I is the only approach that has an explicit term that penalises the model for switching off latents. We have checked empirically whether the different model classes enter the polarised regime across the hyperparameter sweeps reported in our paper, and found that this is indeed the case. All of the models apart from DIP-VAE-I “switch off” on average 3/10 latent dimensions across the hyperparameter sweeps. Saying this, you are also right that the UDR computations do not depend on the models being in the “polarised regime”, so we have re-worded the relevant sections of the paper to avoid confusion.
> > >
> > > In terms of your question about how UDR performs on beta-VAE only, we actually already report UDR performance broken by model class in Figs. 2 and 6 (the latter figure is in the Supplementary Materials). These figures show that UDR ranks beta-VAE models well, both qualitatively and quantitatively, and that its performance for the other model classes is very similar to its performance on beta-VAE.  We have moved some of the results from Fig. 6 to the new Tbl. 1 in the main text. The table reports the correlations between different versions of UDR and the MIG supervised metric across different datasets and model classes. It clearly demonstrates that the different versions of UDR correlate with MIG well and perform comparably regardless of whether they are applied to the beta-VAE or other disentangling VAEs.
> > >
> > > 2) Thank you for suggesting that we test UDR on more naturalistic datasets than the ones presented in the paper. We have just started a hyperparameter sweep training beta-VAE models on ImageNet, Cifar10 and CelebA. These datasets are complex and contain natural images. We will validate UDR rankings qualitatively by comparing the similarity of the visual latent traversals of the best and worst ranked models on these datasets and will report the results of these experiments as soon as they are ready.

---

> > > > ### Author Response · Authors · 2019-11-14
> > > > **Added results on CelebA and ImageNet**
> > > >
> > > > Dear Reviewer,
> > > >
> > > > Our models have finished running on ImageNet and CelebA. We have added a new section in the Supplementary Materials to describe the results and a line to reference these results in the main text. Our results show that despite VAEs being notoriously bad at modelling ImageNet, they were still able to learn how to disentangle the coarse representations achieved on this dataset. The models were able to disentangle CelebA well. UDR ranked the models well on both datasets.
> > > >
> > > > Although we only had time to run these experiments on the beta-VAE, which is the fastest model class to train among the ones considered in this paper, we will be willing to add extra results on the other models classes for the camera ready version of the paper.

---

> > > > > ### Comment · AnonReviewer1 · 2019-11-15
> > > > > **Thank you for your updates.**
> > > > >
> > > > > Thank you for your updates.
> > > > >
> > > > > I still consider the limitations for the UDR is notable and  I might avoid measuring UDR personally because: 1) It is computationally expensive, because it requires training hundreds of disentanglement models. 2) The method is not well theoretically supported. I am still not convinced that UDR will always work as expected, if it is measured on a new dataset, a different set of hyperparameters is used, a different model initialization strategy is used, or another VAE model other than the models mentioned in this paper is involved.
> > > > >
> > > > > However, unsupervised model selection for disentangled representation learning is a difficult problem, and I have not seen any alternative methods in the literature. This might imply that some members of the research community will be interested in this method and the proposed method might motivates them to develop better methods for this problem. Therefore, I have updated my review rating.
> > > > >
> > > > > It looks like the current revision still states that all VAE-based models are likely to enter the "polarised regime" because of the theorems in Rolinek et al. (2019).  I do not agree with the statement and consider it misleading.

---

> > > > > > ### Author Response · Authors · 2019-11-15
> > > > > > **Thank you for changing your score**
> > > > > >
> > > > > > Dear Reviewer,
> > > > > >
> > > > > > Thank you for changing your score. We really appreciate it.
> > > > > >
> > > > > > We have taken your final comments into account and will work to develop further methods for unsupervised disentangled model selection in the future that address them. We hope, however, that in the meantime UDR can be useful for the practitioners using the current disentangling VAEs, and that this paper encourages others to come up with better unsupervised model selection methods.
> > > > > >
> > > > > > In the recent version of the paper we added a paragraph in Sec. 4 (paragraph 4) that discusses whether the assumptions of Rolinek et al hold for other disentangling models apart from beta-VAE, and point to the new section in the Supplementary Materials that discusses this point in more depth. We are not sure if you have seen it. If you believe that we still have misleading statements in our paper after this change, please let us know and we will address them.
> > > > > >
> > > > > > Thank you again for engaging in a meaningful discussion with us.

---

### Official Review · AnonReviewer4 · 2019-11-03
**Official Blind Review #4**

**Rating:** 6

**Review:**

This paper addresses the problem of unsupervised model selection for disentangled representation learning. Based on the understanding of “why VAEs disentangle” [Burgess et al. 2017, Locatello et al. 2018, Mathieu et al. 2019, Rolinek et al. 2019], the authors adopt the assumption that disentangled representations are all alike (up to permutation and sign inverse) while entangled representations are different, and propose UDR method and its variants. Experimental results clearly show that UDR is a good approach for hyperparameter/model selection.
Overall, I think a reliable metric for model selection/evaluation is needed for the VAE-based disentangled representation learning. According to comprehensive experimental studies performed in this paper, UDR seems to be a potentially good choice.

However, I am not sure if very good disentangled representations must benefit (general) subsequent tasks, though the authors provide experimental evidence on fairness classification and data efficiency tasks. Actually, the data generation process in the real-world may consist of different generative factors that are not independent of each other. Though good disentangled representation provides good interpretability, it needs not to be better than entangled representation for concrete tasks. Specifically, for concrete supervised classification tasks, VAE with beta smaller than 1 (not towards disentanglement) might be the best (Alexander A. Alemi et al. 2017, Deep VIB).

Another concern is about the choice of some key “hyperparameters”.
For the KL divergence threshold in equation 3, you set it to be 0.01. It looks like the choice would control how much the UDR favors a “sparse representation map”. The larger the value, the few “informative dimensions” would be considered.
In supplementary material, you say that “uninformative latents typically have KL<<0.01 while informative latents have KL >>0.01”. Is this judgment based on “qualitative feeling”? For me, as you are contributing a ``quantitative measurement”, it is interesting and important to see how this threshold would generally affect UDR’s behavior in one (or more) datasets you have tried.
Another hyperparameter I cared is P (number of models for pairwise comparison). In the paper, you validate the effect of P in the range [5,45]. How would P smaller than 5 affect UDR? According to Table 1, if I was using UDR, I’d rather using P>=20 (or at least 10) rather than 5.
Also, it seems to me P would grow up due to the size of factors that generate the data. Thus, I also have a little concern about the computation cost of the proposed metric (as also mentioned by the authors).

Others concerns:
-- As a heavy experimental paper, most experimental results are in supplementary material, while the authors spent a lot of time in the main text explaining the conclusions found in other papers.
-- To validate the fundamental assumption of UDR, the authors might consider to quantitatively validate that, disentangled representations learned by those approaches you used in the paper are almost the same (up to permutation and sign inverse).

**Experience Assessment:**

I have read many papers in this area.

**Review Assessment: Checking Correctness Of Derivations And Theory:**

I assessed the sensibility of the derivations and theory.

**Review Assessment: Checking Correctness Of Experiments:**

I carefully checked the experiments.

**Review Assessment: Thoroughness In Paper Reading:**

I read the paper thoroughly.

---

> ### Author Response · Authors · 2019-11-08
> **Answer to Reviewer4**
>
> Dear Reviewer,
>
> Thank you for your thoughtful comments. In your feedback you have brought up two points: 1) you were wondering whether disentangled representations would benefit subsequent tasks; 2) you were wondering about our choice of certain hyperparameters. We will address these questions below:
>
> 1) In terms of whether disentangled representations would benefit subsequent tasks, we believe that it is important to consider the nature of the task, and whether it implicitly assumes any of the properties that disentangled representations possess. For example, if one is trying to solve a binary classification task based on the value of a single pixel in a high-dimensional image, it is unlikely that a disentangled representation will be useful. Indeed, a disentangled representation will most likely learn to ignore this pixel, since it doesn’t contribute much to the quality of the reconstruction. On the other hand, an entangled representation learnt implicitly through a supervised objective aiming to solve the task will throw away all information apart from the value of the relevant pixel and hence will be much more informative for that particular task.
>
> On the other hand, if one is interested in solving a large number of natural tasks in a single environment (e.g. learning to achieve different values of the score in an Atari game, generalising policies to variations in the game colour schemes, fast language binding problems, data efficient classification of object identities, colours, sizes or relations), then a disentangled representation may be of more relevance, since it will produce the semantically meaningful equivariant compositional representation that will support many variations of these tasks. Hence, we believe that disentangled representations will be useful for those tasks that require compositionality, generalisation, data efficiency or generalisation/transfer.
>
> You were also wondering whether the real world data generative process follows the independence assumption presumed by disentangled representations. To answer this question we would like to refer you to the recently proposed alternative view on disentangled representations that moves away from considering independent generative factors (Higgins et al, 2018). The new definition suggests that disentangled representations instead reflect the compositional natural symmetry transformations. The implication of this definition is that one can move away from assuming IID training data generated by independent generative factors, and instead think about which aspects of the world can be transformed independently of each other, and how these transformations can be discovered through embodied active learning (see Caselles-Dupre et al, 2019 for a first effort in that direction).
>
> 2) You were wondering about the choice of the 0.01 threshold in Eq. 3, and whether it was set using a “qualitative feeling”. The answer is no. This equation quantifies which latent dimensions are used by the network that has entered a “polarised regime”. Rolinek et al define a model to be in a “polarised regime” if its latents can be split into two disjoint sets of “used” and “unused” dimensions. “Used” dimensions are defined as those which have inferred \sigma^2 << 1, and the “unused” dimensions are defined as those which have \mu^2 << 1 and \sigma^2 \approx 1 (see Sec 3.2, Definition 1). Note that the latter would result in a small KL from a unit Gaussian prior as per Eq. 3. Empirically we found that 0.01 was a good threshold, since the KL values have a bimodal distribution, with on average around 97% of all “small kl” values lying below this threshold.
>
> In terms of P, we do not suggest using P<5. Training P seeds per hyperparameter setting is the largest computational overhead of the UDR, however it is subsumed by the largely accepted good research practice of training a number of seeds per hyperparameter setting anyway. The rest of the UDR computations are very fast. To give you an idea, running a single pairwise comparison using UDR Lasso takes around 4 seconds on a standard CPU, and it takes around 1600 seconds on average to compute UDR Lasso with P=50 for 300 models within a hyperparameter sweep, when we parallelise the pairwise comparisons per model.
>
> Finally, how would you propose that we quantitatively evaluate whether the representations learnt by the models are the same up to the UDR assumptions apart from running the UDR itself? Maybe we misunderstood your question...

---

> > ### Comment · AnonReviewer4 · 2019-11-08
> > **~**
> >
> > For your reply on "whether disentangled representations would benefit subsequent tasks":
> > I am not trying to argue in what cases "disentangled representations" would be more helpful or less. My understanding is the same as what you claimed in the reply, it depends. I basically doubt some of your general claims (e.g. for subsequent tasks). Now I think, most of the results/claims here are based on the context that when we know disentanglement would help.
> >
> > For your reply on "threshold 0.01", now I can buy it.
> > For "p", thanks for detailed information.
> > I am actually not worrying about the speed of the UDR computation. Sorry for misleading you.
> > I agree that UDR itself is not computational intensive.
> > My actual concern is, as P is not very small, we need to train S>P models for one hyper-parameter, which can be expensive. That is why I care about it. To me, it seems that P>=20 (or at least 10) is reasonable, according to Table 1.
> >
> > For your reply on "quantitatively evaluation":
> > I am not saying I don't agree with UDR itself, I am also not asking you to "self-validate" UDR.
> > I am thinking to directly examine some of your trained models to directly support the assumptions.
> > For example, by directly checking J and its SVD, or by directly checking the normalized and ranked features you learned.

---

> > > ### Author Response · Authors · 2019-11-08
> > > **Response**
> > >
> > > Thank you for getting back to us so quickly.
> > >
> > > In terms of P, it is fine to train S=P models per hyper-parameter. Furthermore, if for whatever reason that is not possible, it is also ok to train S<P models per hyper-parameter but run the All-2-All version of UDR where the models for pairwise comparisons are samples across hyperparameter settings. The results of these different versions of UDR are very similar (e.g. see Figs. 6-7 in the Supplementary Materials).
> > >
> > > In terms of the test on J that you are proposing, Rolinek et al actually did something similar for simple beta-VAE models in their paper (https://arxiv.org/pdf/1812.06775.pdf). They proposed a Distance to Orthogonality measure which basically compared how similar the V of the SVD decomposition was to the best matching signed permutation matrix. Unfortunately they required to run Integer Programming to approximate the permutation matrix which is computationally infeasible once the dimensionality of V becomes large, and hence we would struggle to run it directly on our models.
> > >
> > > In terms of a more qualitative test, it is possible to see that the different models learn the same representation (up to a permutation, sign inverse and subsetting) by looking at the latent traversal plots. In these plots we change the value of one latent at a time while fixing the others, and visualise what effect this has on the reconstruction. For example, Fig. 9  in Supplementary Materials shows such latent traversals for a number of beta-VAEs trained during a hyperparameter sweep. Models 1, 2 and 4 in that plot are highly scored by the UDR and it is clear that their representations are the same up to the UDR assumptions. Models 3, 5 and 6 on the other hand are poorly scored according to UDR and their representations look quite different.

---

> > > > ### Comment · AnonReviewer4 · 2019-11-14
> > > > **to author(s)**
> > > >
> > > > Dear Authors,
> > > >
> > > > In your reply, your evidence is mostly about beta-vae and existing findings on beta-vae.
> > > > I still believe that 1) UDR is empirically a good metric according to your comprehensive experiments, 2) the evidence on beta-vae would somehow generalize to some other models, but 3) It is better to pay extra effort to show how your experimented models aligned with your assumptions.
> > > > I think, showing UDR empirically performs well does not directly support your assumptions generally hold for all the approaches you have tried and even further approaches.
> > > >
> > > > If you say "Our method relies on the assumption that for a particular dataset and a VAE-based unsupervised disentangled representation learning model class, disentangled representations are all alike, while every entangled representation is entangled
> > > > in its own way, to rephrase Tolstoy. ", I would definitely suggest some ways (maybe ways you don't prefer) to show how those models empirically match the assumption.
> > > >
> > > > Or, you can circumvent justifying the assumptions. You can say the work is motivated by a sort of observations on beta-vae. You thus design UDR and it turns out to work well for many other models. I can buy that.

---

> > > > > ### Author Response · Authors · 2019-11-14
> > > > > **Added empirical results to verify the assumption**
> > > > >
> > > > > Dear Reviewer,
> > > > >
> > > > > Thank you for further suggestions on how we can improve our paper. We have added a new section to the Supplementary Materials and a few lines to the main text to both empirically verify the assumption, and to make it explicit that the assumption is mainly motivated by the observations on the beta-VAE, but happens to hold for the other model classes too.

---

### Public Comment · ~Cian_Eastwood1 · 2019-10-01
**Comparison of metrics in Table 2**

Interesting paper, I enjoyed reading it. Quick comment on your comparison of disentanglement metrics in Table 2. As you point out in the footnote on page 12, the "modularity", "compactness" and "explicitness" properties of Ridgeway & Mozer (2018) correspond to the "disentanglement", "completeness" and "informativeness" properties of Eastwood & Williams (2018) respectively. However, Table 2 seems to contradict this -- stating that the latter ("DCI" metric) does not in fact measure compactness/completeness or explicitness/informativeness.

---

> ### Author Response · Authors · 2019-10-02
> **Explanation of Table 2**
>
> Thank you for your comment. We followed the notation and supervised metric choices made in Locatello et al (2019) and hence use "DCI Disentanglement" to denote just the "disentanglement" part of the Eastwood and Williams (2018) metric. This is why by this definition "DCI Disentanglement" only measures disentanglement/modularity and does not measure compactness/completeness or explicitness/informativeness. We hope this answers your question.

---

> > ### Public Comment · ~Cian_Eastwood1 · 2019-10-02
> > **Clarification**
> >
> > Ah I see. So Table 2 shows which *disentanglement* metrics are undesirably affected by the other distinct properties, namely compactness/completeness and explicitness/informativeness?

---

> > > ### Author Response · Authors · 2019-10-08
> > > **Further clarification**
> > >
> > > Table 2 shows which of the qualities often assigned to disentangled representations are actually assessed by the different metrics. We do not make a judgement as to which of these are desirable or not and instead leave that to the metric users. However, we hope that the table can help the reader understand why the different metrics sometimes rank the same models slightly differently.

---

### Comment · Area_Chair1 · 2019-11-14
**Reviewers, any further comments during the discussion period?**

Dear Reviewers, thanks for your thoughtful input on this submission, and for your quick replies to the author responses!  If you have additional feedback or questions, it would be great to get them this week while the authors still have time to respond/revise further.

Also, there is a wide range in scores for this submission.  Please consider whether the author responses and/or comments of other reviewers affect your recommendation.  Thanks!

---

### Author Response · Authors · 2019-11-14
**Code to be released on disentanglement_lib soon**

Dear Reviewers,

We wanted to point out that we are currently working with Bachem and Locatello to add UDR to disentanglement_lib (https://github.com/google-research/disentanglement_lib). We were hoping to have the code released by now, but unfortunately the process is taking longer than expected. We hope to have the code open sourced next week.

---

### Decision · Program_Chairs · 2019-12-19

**Decision:**

Accept (Poster)

**Comment:**

The authors address the important and understudied problem of tuning of unsupervised models, in particular variational models for learning disentangled representations.  They propose an unsupervised measure for model selection that correlates well with performance on multiple tasks.  After significant fruitful discussion with the reviewers and resulting revisions, many reviewer concerns have been addressed.  There are some remaining concerns that there may still be a gap in the theoretical basis for the application of the proposed measure to some models, that for different downstream tasks the best model selection criteria may vary, and that the method might be too cumbersome and not quite reliable enough for practitioners to use it broadly.  All of that being said, the reviewers (and I) agree that the approach is sufficiently interesting, and the empirical results sufficiently convincing, to make the paper a good contribution and hopefully motivation for additional methods addressing this problem.